# Revising Yield and Equivalence Factors of Ecological Footprints Based on Land-Use Conversion

**Jincheng Li [1]** **, Xinyue Zhang [2], Xuexiu Chang [1] and Wei Gao [1,*]**

[1]   School of Ecology and Environmental Sciences, Yunnan University, Kunming 650091, China;
      lijincheng1991@126.com (J.L.); changxx@ynu.edu.cn (X.C.)
[2]   School of Agriculture Economics and Rural Development, Renmin University of China, Beijing 100872,
      China; xyz0817@163.com
[*]   Correspondence: gaowei@ynu.edu.cn; Tel.: +86-0871-6741-1402

**Abstract:** In the current ecological footprint *(EF)* calculations, the parameters of built-up land are set as equal to those of cropland, based on the assumption that built-up land is totally converted from cropland. However, built-up land may be derived from other types of land use. With the expansion of built-up area as a result of urbanization, the yield and equivalence factors of built-up land are becoming increasingly important in the *EF* calculation. The objective of this study was to evaluate the influence of this assumption on *EF* calculation. In this study, the proportions of different land types converted into built-up land were evaluated based on actual land-use conversion in two urbanized areas of Yunnan Province and Kunming City from 1980 to 2010 in the ArcGIS platform. Then, the parameters of built-up land were calculated by an area-weighting approach with the proportions. The results showed the following: (1) In both cases, the *EF* of Yunnan Province and Kunming City were greater than their biocapacities (*BC*), indicating that they were in unsustainable states. (2) The *EF* and *BC* of the two studied cases were reduced to varying degrees because the yield and equivalence factors of built-up land from land-use conversion are less than cropland factors. As the proportion of the built-up land area in Kunming City was larger than that in Yunnan Province, the reduced proportion of the *EF* and *BC* of Kunming City is greater than that of Yunnan Province. (3) The proportion of built-up land converted from cropland has a significantly positive correlation with *EF* and *BC*. Therefore, it is of great significance to revise the yield and equivalence factors of built-up land using actual land-use conversions in highly urbanized areas.

**Keywords:** built-up land; cropland; yield factor; equivalence factor; land use; conversion

## 1. Introduction

With the continuous expansion of the global population and resource consumption, pollution and other human activities are increasingly close to their environmental capacity, and are exceeding it in some regions, which seriously degrades human sustainability [1]. Since 1987, sustainability has become the development strategy of countries all over the world [2]. Quantifying sustainability and transforming it into an operational model is vital for management [3]. Therefore, many international organizations and related researchers have explored methods and indicators to quantify sustainability, such as the National Wealth Indicator System [4], Sustainable Economic Welfare Index [5], and Barometer Sustainability [6]. However, evaluation methods based on these index systems focus on the assessments of social economic development status, and there is still a large degree of subjectivity and uncertainty in the selection of evaluation indicators, weight determination, and index calculation methods. How to quantify sustainability has long been a difficult problem in sustainable development research. William Rees proposed the ecological footprint (*EF*) concept, which quantifies sustainability

through two normalized indicators to determine human survival within the scope of ecosystem capacity [7]. The *EF* has been demonstrated to be a good approach for measuring ecological impact and capacity [8–10].

The *EF* is primarily used to calculate the bioproductive land of a region that is necessary for the maintenance of resource consumption and the absorption of generated waste by human activities [11]. The *EF* determines (1) the area of cropland, pasture, forest, and fishing ground used for the production of agriculture, livestock, forest, and fishery, respectively, that are consumed by humans, and (2) the forest area for absorbing carbon released by burning fossil fuels [12]. The *EF* is compared with the area's available biocapacity (*BC*). By comparing the *EF* and *BC*, it can be determined whether a region is sustainable. When the *EF* is larger than the *BC*, it is considered to be an "ecological deficit", indicating that human beings are in an unsustainable state. By contrast, if the *EF* is smaller than the *BC*, humans are considered to be in a sustainable state [1,13–16]. To quantitatively research the regenerative capacity needed to maintain a given resource stream, the calculation of the *EF* is based on some assumptions [17]. These assumptions are also a prerequisite to the wide application of the *EF* approach. However, there are many controversies surrounding these assumptions that result in uncertainties being brought into the calculations, and these assumptions are worthy of further assessment [12,18–21]. In the process of an *EF* calculation, it is assumed that the built-up land is all converted from cropland, so the yield and equivalence factors of cropland are adopted for the built-up land. However, in reality, built-up land can be transformed from other types of land such as forest, pasture, and fishing ground. Since the yield and equivalence factors vary substantially among different land uses, and the factors of cropland usually rank the highest, the assumption may overestimate the *EF* and *BC* of built-up land. How does built-up land impact the *EF* and *BC*? At different research scales, is there any difference in this kind of impact? The answers to these questions are still largely unknown. Therefore, the aim of this study was to assess the impact of this hypothesis in order to improve the accounting method for calculating the *EF* and *BC*.

Based on the integrity and accessibility of data, we chose Yunnan Province and Kunming City—which have different research scales and various built-up land percentages—as the case studies, and calculated their *EF* and *BC* in 2010. Using historical land-use data, a land-use conversion matrix was established to calculate the sources of built-up land from 1980 to 2010 through the ArcGIS platform. Then, the yield and equivalence factors of the built-up land were calculated from those factors based on the proportion of its sources. Comparing the revised and unrevised *EF* and *BC*, the changes in the *EF* and *BC* under different built-up land factors were analyzed, and the impact of the assumptions for *EF* and *BC* calculations were explored. Furthermore, the impact of the assumptions was explored under different spatial scales by comparing the changes in Yunnan Province and Kunming City.

## 2. Materials and Methods

### 2.1. Study Area

Yunnan Province is located at 21°8′–29°15′ N and 97°31′–106°11′ E in southwestern China, and Kunming is the capital of Yunnan Province, located at 102°10′–103°40′ E and 24°23′–26°22′ N. In 2010, the area of Yunnan Province was $3.83 \times 10^5$ km$^2$, with a population of 46.01 million, of which, the urban population accounted for 35.20%, and the rural population accounted for 64.80%. The area of Kunming was $2.15 \times 10^4$ km$^2$, with a population of 6.36 million, of which the urban population accounted for 64.00%, and the rural population accounted for 36.00%. Forest, cropland, and pasture are the dominant land areas in Yunnan Province (Figure 1). From 1980 to 2010, the area of cropland, pasture, unutilized land, and fishing ground in Yunnan Province decreased, while forest and built-up land areas increased continuously. However, the built-up land in Kunming increased significantly, which was mainly due to its rapid urbanization [22].

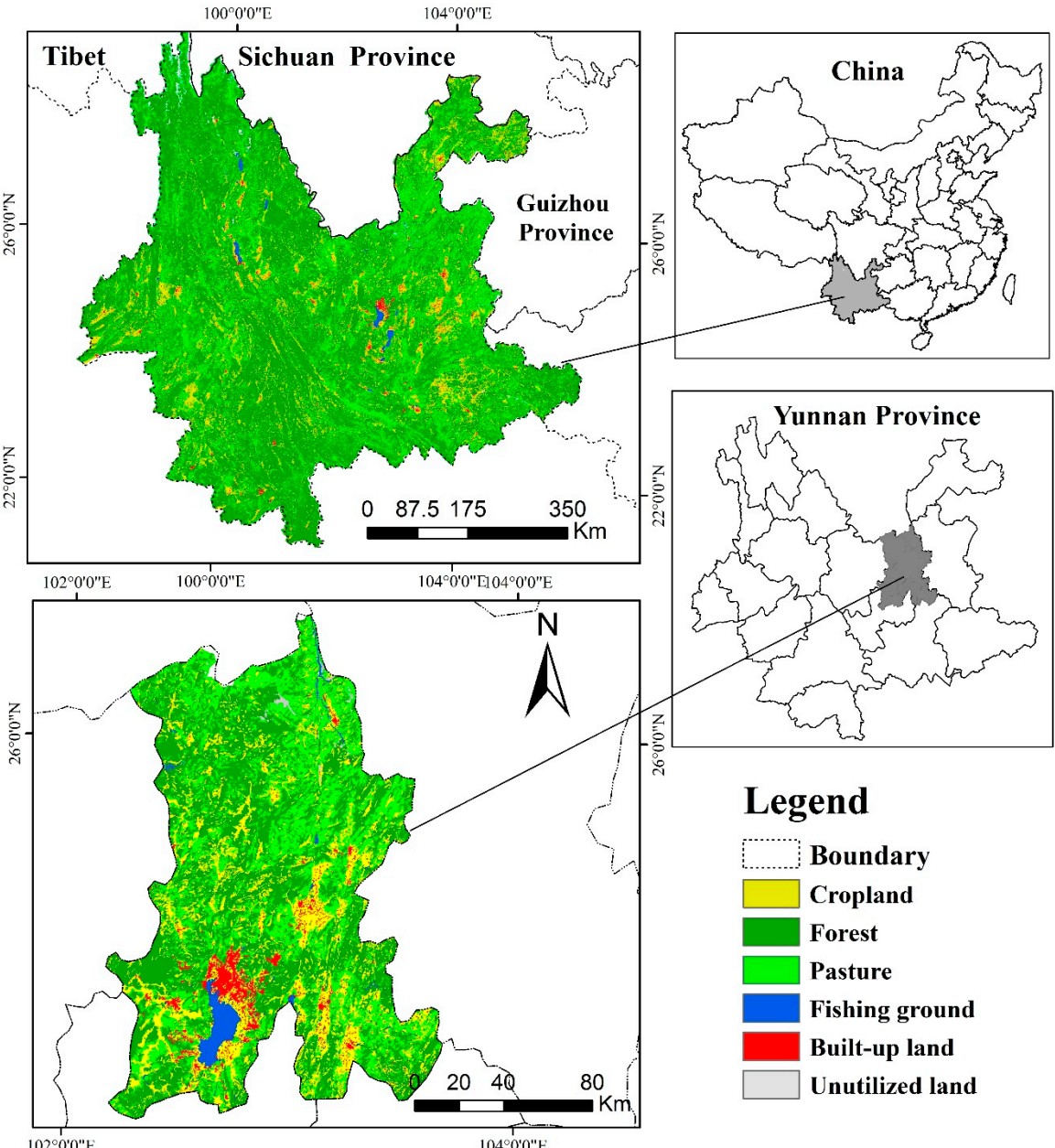

**Figure 1.** Location and land use of Yunnan Province and Kunming City in 2010.

Yunnan Province plays an important role in maintaining global biodiversity and provides a natural defense for southwestern China [23]. As urbanization accelerated in this region, most of its cropland has gradually been converted into built-up land, followed by a large area of forest and pasture that converted to cropland [22]. Yunnan Province acts as a good case for studying the relationship between environmental change and human impacts.

*2.2. Methods*

2.2.1. The *EF* Model

The *EF* was developed by William Rees in 1992, and perfected by Wackernagel in 1996, to measure the human use of natural resources and the functions of natural support service for human. It measures the sustainability of a region by estimating the amount of mutually exclusively bioproductive area

required to sustain human natural resource consumption and assimilate human-generated waste, and comparing it with the biocapacity of this region [7–10].

The *EF* model is used to calculate the area of bioproductive land that is necessary to sustain human resource consumption and waste absorption in the region. In the *EF* model, human resources and energy consumption items are expressed as 5 kinds of bioproductive lands: cropland, pasture, forest, built-up land, and fossil fuel land. Since the bioproductivity varies between different land-use types and between different regions, the *EF* is expressed in terms of the world average bioproduction area (i.e., global hectares (gha)), calculated by multiplying the yield and equivalence factors. In this way, comparisons can be made between different regions and land uses [16]. The *EF* can be calculated as follows:

$$EF = N \cdot \sum_i \frac{C_i}{Y_{ni}} \cdot YF_i \cdot EQF_i = \sum_i \frac{P_i}{Y_{wi}} \cdot EQF_i = EF_b + EF_c + EF_p \,, \tag{1}$$

where *i* is the set of six land-use types; *N* is the total population; $C_i$ is per capita consumption of *i*; $P_i$ is the total consumption of i; $Y_{ni}$ is the yield of *i* in the region; $Y_{wi}$ is the average yield of *i* in the world; $YF_i$ is the yield factor of *i*; $EQF_i$ is the equivalence factor of *i*; $EF_b$ is the *EF* of biological resources; $EF_c$ is the *EF* of fossil fuel-carbon; and $EF_p$ is the *EF* of built-up land.

The yield factors (*YFs*) are the ratios of the yield of the bioproductive areas in a region (or a country) to the global average yield of the same species. They show the differences in biological productivity between regions, and might reflect natural situations, such as temperature or precipitation, as well as anthropogenic management levels [13,24]. The yield factors can be estimated from the actual agricultural production or the net primary productivity (NPP) [19,25–27]. In this study, we used the ratios of Yunnan's NPP to the global average NPP to represent the yield factors of different bioproductive areas in Yunnan Province. The NPP of different bioproductive areas in Yunnan were derived from Liu and Li [28], and the resulting yield factors were 1.62, 1.13, 1.50, and 2.18 for the cropland, forest, pasture, and fishing ground, respectively.

The equivalence factors (*EQFs*) represent the global average potential productivity for a given bioproductive area relative to the average potential productivity of all global production areas. Since the potential productivity of different bioproductive areas is different [24,25], it is necessary to multiply the equivalence factors to transform them into global hectares for the convenience of comparison. Similar to the yield factors, the equivalence factors can be estimated from the NPP or emergy [29]. The potential productivity of different bioproductive areas was not the focus of this study, so we chose the results of World Wildlife Fund (WWF) in 2002 [30]. The equivalence factor of the cropland, forest, pasture, and fishing ground were 2.19, 1.38, 0.48, and 0.36, respectively. Following other studies, the equivalence factor of fossil fuels in this paper was the same as that of forest [24,26,31]. The equivalence factor of built-up land is usually set the same as cropland land under the assumption that built-up land is totally derived from cropland. In this study, an area-weighted equivalence factor based on the actual sources of built-up land was used by calculating the land-use conversion from 1980 to 2010 to the built-up land in the cases studied.

The *EF* of Biological Resources

Cropland, forest, pasture, and fishing ground provide biorenewable resources for human economy and society. The footprint of each land use is the sum of all the consumed products within the category. Table 1 shows the contents of the products included in each category. The average global yield of each product comes from the study of Wackernagel et al. [31]. The formula for the *EF* of each biological area is as follows:

$$EF_b = \sum_i \frac{Q_{cap} \cdot C_{per\,cap}}{Y_{ga}}, \tag{2}$$

where *i* is the kind of bioproductive area; $Q_{cap}$ is the population; $C_{per\,cap}$ is the consumption of per person; and $Y_{ga}$ is the global average yield of the bioproductive area.

<div align="center">**Table 1.** The kind of products included in each area.</div>

| Components | Items | Biological Resources and Energy Types |
|---|---|---|
| Biological resources | Cropland | cereals, veg and fruit, sugar, edible vegetable oil, egg, pork, poultry |
| | Pasture | bovine, goat, mutton and buffalo meat, milk |
| | Forest | fire wood, sawn wood, wood-based panels, wood pulp, paper and paperboard |
| | Fishing ground | aquatic products |
| | Built-up land | buildings |
| Carbon footprint | Fossil fuels | coal, fossil gas, liquid fossil fuel |

The *EF* of Fossil Fuel-Carbon

As a result of fossil fuel combustion, greenhouse gases (mainly carbon dioxide, $CO_2$) have posed a great threat to global sustainability, and have been considered to be important to the *EF*. The $CO_2$ footprint estimates the additional bioproductive area that can absorb $CO_2$. Since $CO_2$ is mainly absorbed by forests in terrestrial ecosystems, fossil energy lands are treated as forests. In addition, the oceans are able to absorb carbon dioxide and contribute a significant percent of their absorption. It has been reported that a quarter of the total anthropogenic emissions should be deducted from the calculation [32]. The formula for the *EF* of fossil fuels ($EF_c$) can be presented as follows:

$$EF_c = P_c \cdot \frac{(1 - S_{ocean})}{Y_c} \cdot EQF_f, \tag{3}$$

$$P_c = P_{bc} + P_{oc} + P_{ng}, \tag{4}$$

where $P_c$ is the total $CO_2$ of anthropogenic emissions; $S_{ocean}$ is the proportion of $CO_2$ absorbed by oceans (1/4); $Y_c$ is 1.42 t/ha, which is the global average rate of $CO_2$ absorbed per hectare of forest [33]; $EQF_f$ is the equivalence factor of the forest; $P_{bc}$ is the carbon dioxide emitted from burning coal; $P_{oc}$ is the carbon dioxide emitted from burning oil; and $P_{ng}$ is the carbon dioxide emitted from burning natural gas.

According to the United States Oak Ridge National Laboratory [34], the carbon dioxide generated from burning coal could be calculated as follows:

$$P_{bc} = M_c \cdot EOF \cdot C_{pc}, \tag{5}$$

where $M_c$ is the standard coal equivalent of burning coal; *EOF* is the effective oxidation fraction (*EOF* = 0.98); and $C_{pc}$ is the carbon content of per ton of standard coal ($C_{pc}$ = 0.73).

$$P_{oc} = M_o \cdot EOF \cdot C_{pc}, \tag{6}$$

where $M_o$ is the standard coal equivalent of oil consumption.

$$P_{ng} = M_{ng} \cdot EOF \cdot C_{pc}, \tag{7}$$

where $M_{ng}$ is the standard coal equivalent of natural gas.

In Yunnan Province, there is no nuclear power station and little biomass substitution, so they are not included here.

The EF of Built-Up Land

In reality, built-up land may be derived from the other four types of land uses, so the *YF* and *EQF* of the built-up land should be equal to the area-weighted YFs and EQFs of the land uses the built-up land replaces, rather than cropland alone. According to the land uses in 1980 and 2010, the weight

coefficient of the area can be obtained through the transfer matrix analyses in the ArcGIS platform. In this study, the *EQF* and *YF* of the built-up land were calculated from the perspective of the actual land-use conversion. Thus, the EF of built-up land can be presented as follows:

$$EF_p = A_p \cdot EQF_p \cdot YF_p, \tag{8}$$

where $A_p$ is the area of built-up land; $EQF_p$ is the equivalence factor of built-up land; and $YF_p$ is the yield factor of built-up land.

$$EQF_p = R_f \cdot E_f + R_p \cdot E_p + R_c \cdot E_c + R_w \cdot E_w, \tag{9}$$

where $R_f$ is the area ratio of forest to built-up land; $E_f$ is the equivalence factor of forest; $R_p$ is the area ratio of pasture to built-up land; $E_p$ is the equivalence factor of pasture; $R_c$ is the area ratio of cropland to built-up land; $E_c$ is the equivalence factor of cropland; $R_w$ is the area ratio of fishing ground to built-up land; and $E_w$ is the equivalence factor of fishing ground.

$$YF_p = R_f \cdot Y_f + R_p \cdot Y_p + R_c \cdot Y_c + R_w \cdot Y_w, \tag{10}$$

where $R_f$ is the area ratio of forest to built-up land; $Y_f$ is the yield factor of forest; $R_p$ is the area ratio of pasture to built-up land; $Y_p$ is the yield factor of pasture; $R_c$ is the area ratio of cropland to built-up land; $Y_c$ is the yield factor of cropland; $R_w$ is the area ratio of fishing ground to built-up land; and $Y_w$ is the yield factor of fishing ground.

Calculation of the *BC*

The *BC* is the counterpart of the *EF*, indicating the strength of the carrying capacity for a region. A nation's *BC* is the sum of all its bioproductive areas and is usually described in global hectares. Each bioproductive area was transformed into global hectares by multiplying its *EQF* and *YF*.

$$BC = \Sigma A_i \cdot EQF_i \cdot YF_i, \tag{11}$$

where *i* is the set of five land-use types; $A_i$ is the area of *i*; $EQF_i$ is the equivalence factor of *i*; and $YF_i$ is the yield factor of *i*.

Calculation of the *ED*

The relative magnitudes of *EF* and *BC* reveal whether existing natural resources are sufficient to support social development and human consumption. If the *BC* is larger than the *EF*, then there is ecological reserve (*ER*). If the *EF* is larger than the *BC*, then there is ecological deficit (*ED*), indicating a mismatch occurring locally in the region [35].

$$ED = EF - BC \tag{12}$$

*2.3. Data Sources*

Yunnan Province and its capital, Kunming City, were selected as the two cases studied in this study, considering their varying urbanization levels. The basic socioeconomic data (population, GDP, PGDP, energy resource consumption, commerce) used in this study were derived from the Statistical Yearbook of Yunnan Province in 2011, and the County's Statistical Yearbook of Yunnan Province in 2011. The data of production and consumption of biological resources used in this study were derived from the China Urban Statistical Yearbook 2011 and the China Rural Statistical Yearbook 2011.

The land-use data was obtained from Geographical Information Monitoring Cloud Platform (http://www.dsac.cn/DataProduct/Detail/200804), with a spatial resolution of 30 m, and the data were acquired in 1980 and 2010, respectively. In this study, land-use types were divided into six

categories, and the classification methods referred to China National Current Land-Use Classification Standard (GB/T 21010-2017) [36]. The specific classification which was used in this study has been published in many studies [37–39]. The land-use conversion data were obtained in the ArcGIS platform. The method was used in some publications [40–42]. In the ArcGIS platform, firstly, the land-use data of 1980 and 2010 were merged (the "Dissolve" function); secondly, layer overlay analyses were performed (the "Intersect" function); thirdly, a new attribute list (the "Calculate Geometry" function) was established; and, fourthly, matrix analyses were performed for the obtained data through Excel2016. Finally, the land-use conversion data were obtained from 1980 to 2010.

## 3. Results and Discussion

### 3.1. The Factors of Built-Up Land

The land-use conversion data were obtained through the matrix analyses transfer of the land use between 1980 and 2010. The built-up land area of Yunnan Province was 1732.61 km$^2$ in 1980, and increased to 3083.98 km$^2$ in 2010. The area of built-up land increased by 1351.36 km$^2$ in 30 years. The area converted from cropland to built-up land was 1103.68 km$^2$, which accounted for 73.05% of the total conversion area. The changes are shown in Figure 2. Therefore, cropland was the main source of converted lands for the built-up area. This result has some similarities with the assumption of the built-up land footprint. However, it is obvious that other land uses also contributed significantly. The area converted from forest to built-up land was 179.29 km$^2$, which accounted for 11.88% of the area converted. The area converted from pasture to built-up land was 211.36 km$^2$, making up 14.20% of the total converted area, and the area converted from fishing ground to built-up land was 12.86 km$^2$, which made up the remaining converted area. These parts of the total conversion area accounted for 26.95% of the total area converted, which is different from the assumption of the construction land footprint (Figure 3).

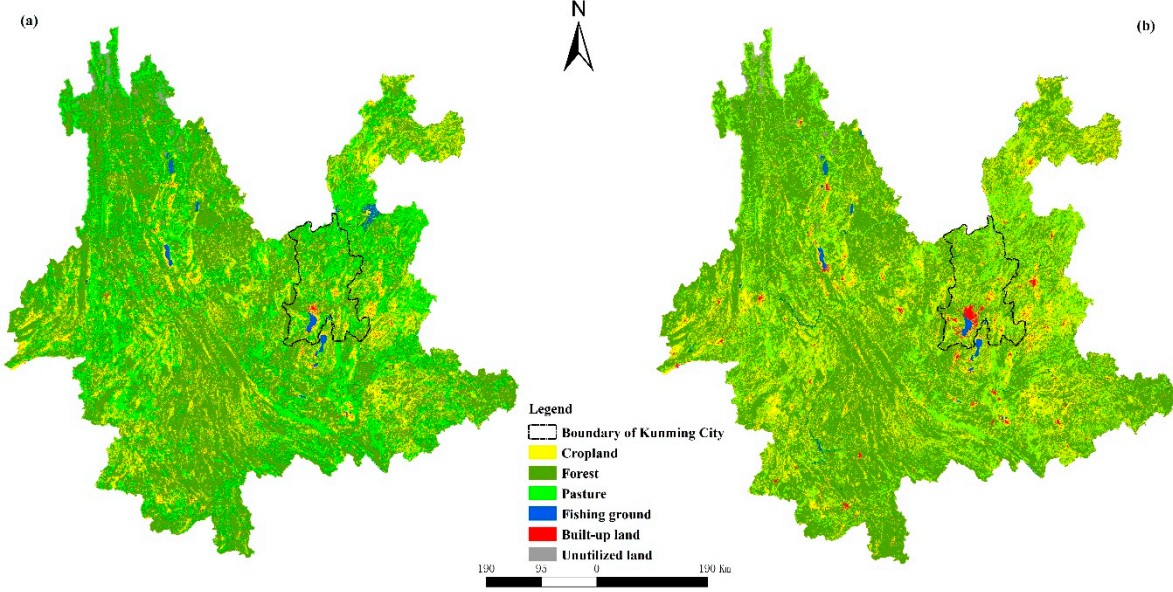

**Figure 2.** Land use of Yunnan Province and Kunming City in (**a**) 1980 and (**b**) 2010.

The results of the land-use transformation matrix in Kunming City from 1980 to 2010 were different from those in Yunnan Province. The area of built-up land increased by 293.62 km$^2$ in 30 years. The area had a total conversion area of 64.95% that was converted from cropland to built-up land. The total area converted from forest to built-up land was 12.11%. The area that was converted from pasture to built-up land was 22.94%. The changes were shown in Figure 3. The cropland was the main source of converted lands for built-up areas, but the other land uses clearly contributed. The proportion of

built-up land originating from cropland in Yunnan Province was larger than that in Kunming City. This difference was relative to the level of economic development, urbanization rate, land-use policies, and so forth [43–47]. In 2010, the urbanization rate of Yunnan Province was 35.20%, and the per capita GDP was 15,752 RMB. The urbanization rate of Kunming City was 64.00%, and the per capita GDP was 33,549 RMB. As of 31 December 2010, there were total 93 laws and regulations, related to land transfer, management, and land-use planning, had been shown on the official website of Kunming Municipal Bureau and Yunnan Provincial Department of Land and Resources. This factor was one of the main reasons that led to the difference in the land-use change.

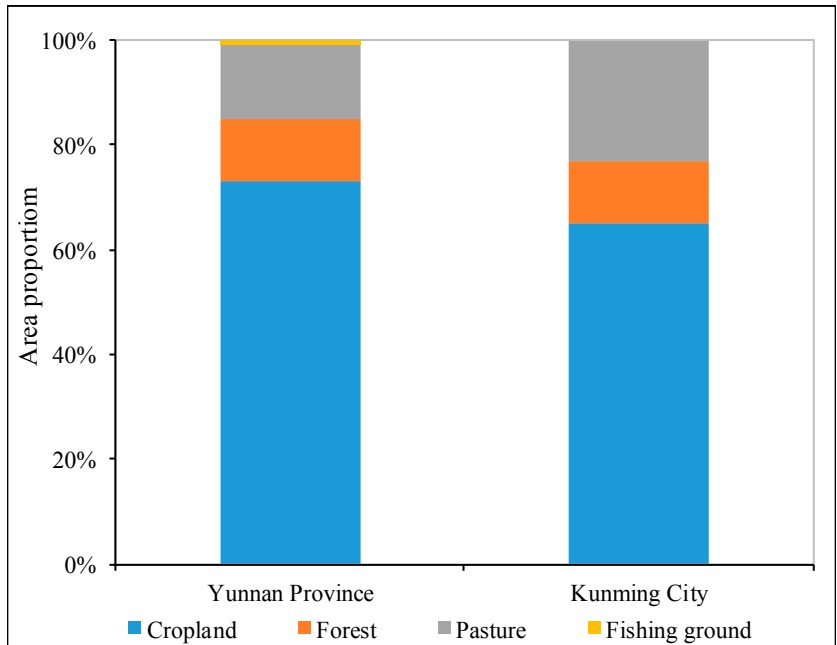

**Figure 3.** The conversion source for built-up land in Yunnan Province and Kunming City between 1980 and 2010.

According to the results of the land-use conversion matrix and the yield factors of the four kinds of land use, the yield factors of built-up land were weighted according to the conversion ratio. The revised yield and equivalence factors of built-up land in Yunnan Province were 1.55 and 1.84, respectively (Figure 4). The revised yield and equivalence factors of built-up land in Kunming City were 1.53 and 1.70, respectively. It is obvious that there are some differences in the yield and equivalence factors under different research scales. In general, the revised yield and equivalence factors were smaller than those without the revision.

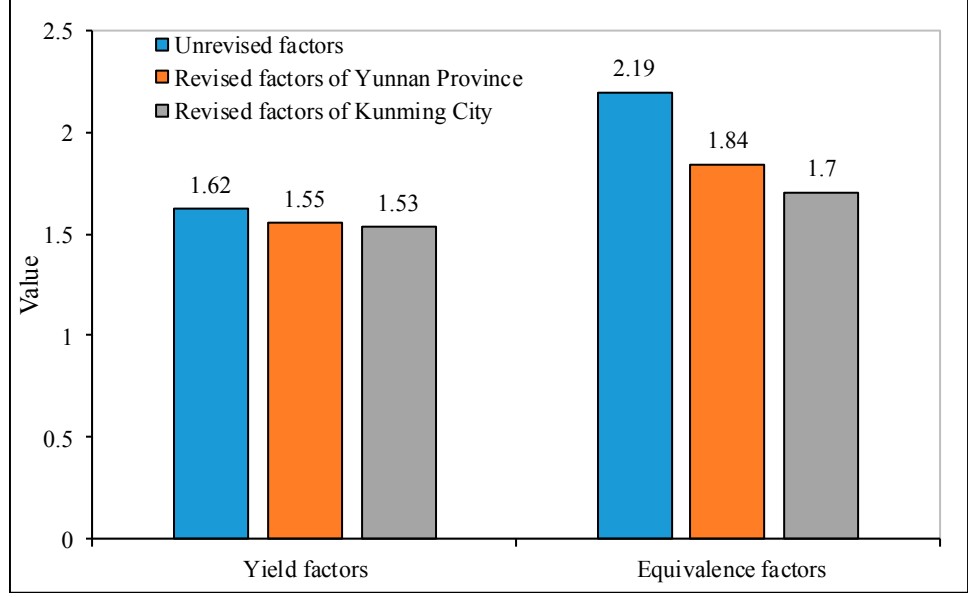

**Figure 4.** The yield and equivalence factors of built-up land in Yunnan Province and Kunming City.

*3.2. Case Study: Yunnan Province*

3.2.1. Estimating *EF*

According to the classified catalog in Table 1, we obtained the *EF* of cropland, forest, pasture, and fishing ground as 26.99 million gha, 20.09 million gha, 4.58 million gha, and 1.14 million gha, respectively. Based on the Statistical Yearbook of Yunnan Province in 2011 and the Statistical Yearbook of China in 2011, the fossil energy consumption in Yunnan Province is shown in Table 2. The total fossil energy consumption was 62.80 million tons of standard coal, and the area of fossil energy land obtained through the carbon dioxide footprint calculation formula was 34.13 million gha.

**Table 2.** The ecological footprint (*EF*) of fossil fuel-carbon.

| Category | Amount Consumed (t) | Carbon Emissions (t) | Fossil Energy Land (gha) |
|---|---|---|---|
| Coal * | 49,208,566.41 | 35,203,808.41 | 28,994,798.68 |
| Liquid fossil fuel * | 13,106,670.87 | 9,376,512.34 | 4,952,383.28 |
| Fossil gas * | 485,753.52 | 347,508.07 | 183,542.99 |
| Total | 62,800,990.8 | 44,927,828.82 | 34,130,724.95 |

\* The amount of consumption of fossil fuel has been transformed to standard coal.

Based on the *EF* of built-up land formula, we obtained the *EF* of built-up land as 0.88 million gha with revised factors, and the *EF* of built-up land was 1.08 million gha with unrevised factors. Hence, a 19.61% reduction in the *EF* of built-up land was seen with the use of revised factors.

The *EF* of Yunnan Province is shown in Figure 5. With the revised factors, the *EF* of Yunnan Province was 88.03 million gha. Cropland, forest, fossil fuel-carbon, pasture, fishing ground, and built-up land accounted for 30.74%, 22.88%, 38.87%, 5.22%, 1.30%, and 1.00%, respectively, of the *EF*. This phenomenon is similar to the *EF* of China from 1978 to 2010 [43]. Since China is an agricultural country, its agricultural production and consumption are higher than those of other products. However, the *EF* of fossil fuel-carbon has become the largest component in the consumption footprint since 2005, which increased to 35% in 2010. The similar proportion of fossil fuel-carbon indicates that the fossil fuel consumption of Yunnan Province has reached the average of China. The difference is that the proportion of pasture land in Yunnan Province is less than that in China, because the production

and consumption of pasture land in Yunnan Province is scarce. Based on these proportions, the *EF* proportions of cropland, forest, and fossil fuel-carbon were the main components as they accounted for 92.48% of the *EF*. This result showed that the consumption of resources and energy was the main source of the *EF*. However, the ratio of fossil fuel-carbon was the largest, which indicated that economic development depended on fossil fuel and the energy consumption accounted for a large proportion of economic growth. Compared with the research of the Ecological Footprint Dynamics of Yunnan, in this study, the proportion of fossil energy land in the total *EF* increased continuously from 1988 to 2006. Specifically after 2000, the growth rate was obvious and, in 2006, the *EF* of fossil energy land accounted for 35% of the total *EF* in Yunnan Province [48]. Therefore, energy is likely to be the main factor restricting the sustainability of Yunnan Province in terms of social development. This phenomenon is similar to those in past publications [29,49–52]. With the unrevised factors, the *EF* of Yunnan Province was 87.82 million gha. Among them, the *EF* of cropland, forest, fossil fuel-carbon, pasture, fishing ground, and built-up land separately accounted for 30.66%, 22.82%, 38.77%, 5.20%, 1.30%, and 1.24%, respectively. The proportion was similar to that with the revision, but the proportion of built-up land is slightly increased. From the results, we can determine that the revised *EF* of Yunnan Province was 0.22 million gha less than that without the revision, which made up the unrevised *EF* of 0.23%. The revised *EF* per capita was 0.01 gha less than that without the revision. This result indicates that the revised factors lead to a reduction in the *EF*, but the reduction is small in proportion to the total amount, especially to the average person, with little influence.

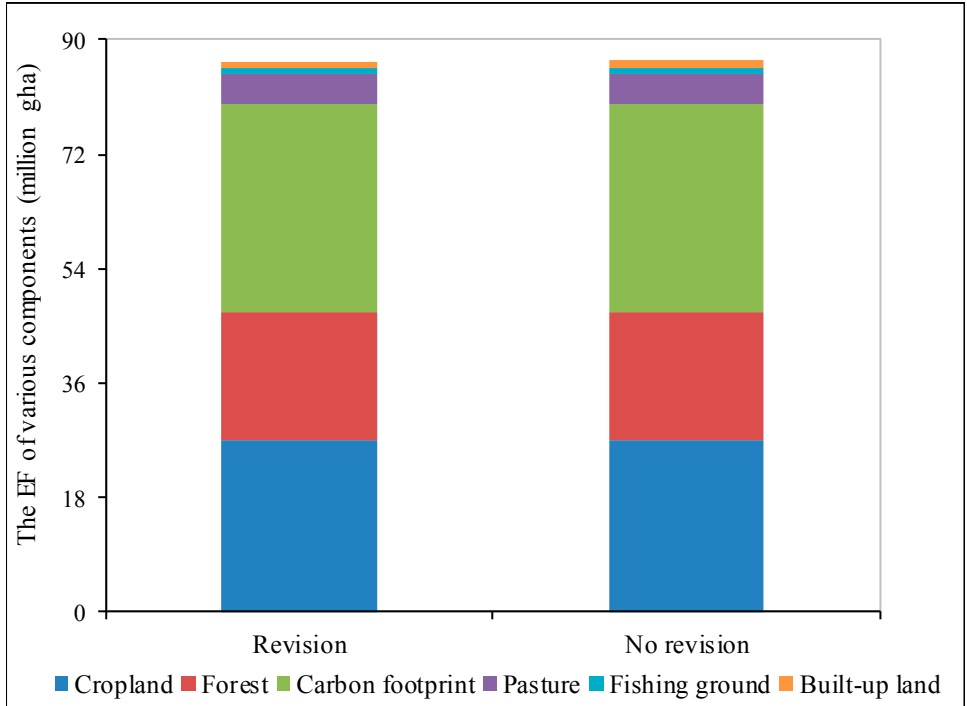

**Figure 5.** The *EF* of various components with revision and with no revision.

### 3.2.2. Estimating *BC*

The results of the *BC* calculations are shown in Table 3. With the revised factors, the *BC* of built-up land was 0.88 million gha, and the total *BC* was 66.02 million gha. Cropland, forest, pasture, fishing ground, and built-up land accounted for 36.84%, 52.03%, 9.45%, 0.34%, and 1.33% of the *BC*, respectively. The largest proportion was forest, indicating that the forest provided abundant forest products and absorbed much of the waste gases in Yunnan Province. With the unrevised factors, the *BC* of built-up land was 1.09 million gha, and the total *BC* was 66.24 million gha. Compared with the *BC* of built-up land, the revised *BC* was 0.22 million gha less than that without the revision, which

accounted for 19.61% of the unrevised *BC* of built-up land. The total *BC* was 0.22 million gha less than that without the revision, which accounted for 0.32% of the unrevised *BC*. The *BC* per capita was 0.01 gha less than that without the revision. The results showed that the revised factors had a great influence on the *BC* of built-up land, a small influence on the total *BC* and little effect on the per capita *BC*. The result is similar to the influence of the revised factors on the *EF*.

**Table 3.** The biocapacity (*BC*) of Yunnan under different assumptions

| Land Use | Area (ha) | Revision (gha) | No Revision (gha) |
|---|---|---|---|
| Cropland | 6,855,811.11 | 24,323,046.67 | 24,323,046.67 |
| Forest | 22,029,953.08 | 34,353,508.84 | 34,353,508.84 |
| Pasture | 8,669,372.65 | 6,241,948.31 | 6,241,948.31 |
| Fishing ground | 287,044.18 | 225,272.27 | 225,272.27 |
| Built-up land | 308,361.78 | 879,447.78 | 1,094,005.91 |

According to the *ED* formula, we obtained a total *ED* of 21.80 million gha, and the revisions have no effect on the ED, which is 0.48 gha per capita. Since the *EF* and *BC* both decreased, and the decreased amount was identical after the revised factors, the revised factors had no effect on the *ED*. The *ED* of Yunnan Province was larger than that in this study [48]. The reason for this result is that we used different yield and equivalence factors.

### 3.3. Case Study: Kunming City

#### 3.3.1. Estimating *EF*

In 2010, the population of Kunming City was 6.39 million people, of which the rural population was 2.32 million, and the urban population was 4.07 million. On the basis of the Statistical Yearbook of Kunming City in 2011, the Yunnan Province Statistical Yearbook in 2011, and survey data, the consumption of various types of biological products were obtained through calculation. The *EF* of cropland was 2.68 million gha, the *EF* of forest was 1.67 million gha, the *EF* of pasture was 2.31 million gha, and the *EF* of fishing ground was 4.28 million gha. The total fossil energy consumption was 17.72 million tons of standard coal, and the area of fossil energy land obtained through the carbon dioxide footprint calculation formula was 10.44 million gha. The *EF* calculation method of built-up land was consistent with the provincial level. The unrevised *EF* of built-up land was 0.24 million gha, and the revised *EF* of built-up land was 0.18 million gha—a 26.55% reduction in the *EF* of built-up land using revised factors.

The *EF* of Kunming City is shown in Figure 6. The revised *EF* was 21.56 million gha. The cropland, forest, fossil fuel-carbon, pasture, fishing ground, and built-up land accounted for 12.43%, 7.75%, 48.43%, 10.70%, 19.86%, and 0.83%, respectively, of the *EF*. Based on these proportions, the ratio of fossil fuel-carbon was the largest, which was similar to that of Yunnan Province, but the proportion of fossil fuel-carbon was much higher than that of Yunnan Province. This result shows that the economic development of Kunming City relied more on fossil energy than that of Yunnan Province. The proportion of fishing ground was very high, and was larger than that of Yunnan Province. This result is strongly related to Kunming's industrial structure. In 2010, the proportion of tertiary industry in Kunming City accounted for 49.01% of GDP, larger than that in Yunnan Province, which accounts for 40.04% of GDP. In the tertiary industries, tourism provided the largest contribution. This led to the proportion of fishing ground of *EF* in Kunming City being larger than that in Yunnan Province. Factors, such as urbanization levels and per capita GDP, also contribute to these differences. The unrevised *EF* was 21.62 million gha, and the proportions of the various components were similar to those that were revised. Therefore, the total *EF* was reduced by 64,830 gha, which was 0.28% of the unrevised *EF*. The *EF* per capita was reduced by 0.01 gha, which was minimally changed.

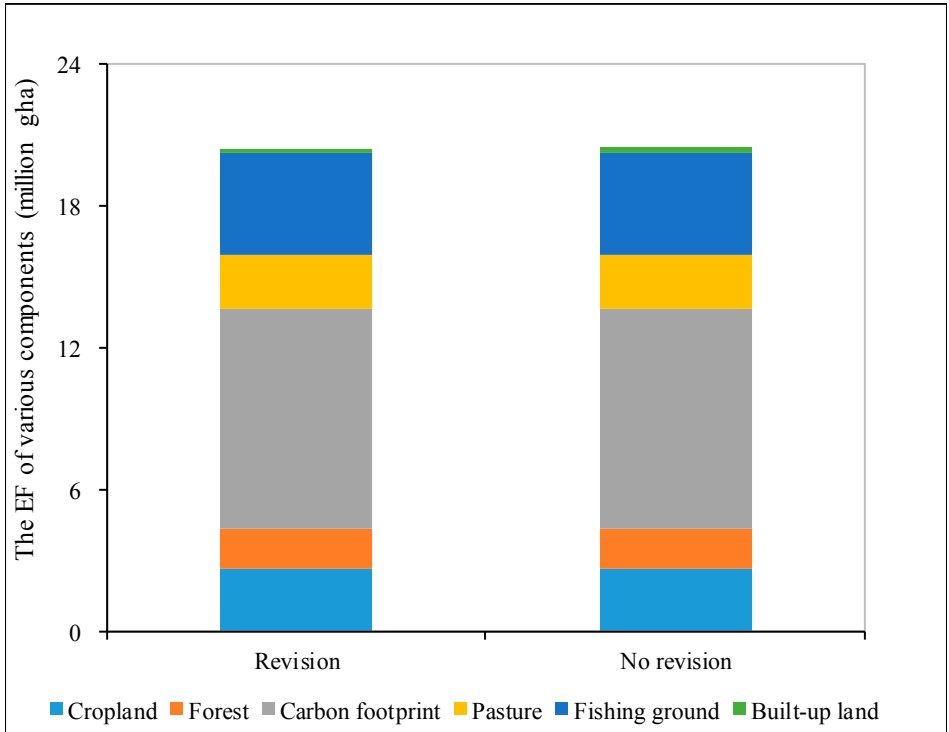

**Figure 6.** The *EF* of various components with revision and no revision.

### 3.3.2. Estimating *BC*

With the revised factors, the *BC* of built-up land was 0.18 million gha, and the total *BC* was 3.64 million gha. Cropland, forest, pasture, fishing ground, and built-up land accounted for 40.44%, 40.34%, 11.60%, 1.03%, and 6.59% of the *BC*, respectively. The largest proportions were cropland and forest, indicating that cropland resource was a primary element of the local biocapacity. Compared with Wuhan City, the largest proportion of the biocapacity was cropland, which exceeded 80%, and the proportions of forest and pasture were lower than 5% [35]. This indicated that the biocapacity of forest and pasture in Kunming City had a larger contribution to the total biocapacity than that in Wuhan City. This phenomenon is the same as the development goals of two cities in China. The main goal of Wuhan is to develop heavy industry. For Kunming, the main goal of future urban development is to develop tourism based on the protection of the environment. With the unrevised factors, the *BC* of built-up land was 0.24 million gha, and the total *BC* was 3.70 million gha. Compared with the *BC* of built-up land, the *BC* was reduced by 64.8 thousand gha, which accounted for the *BC* of built-up land with an assumption of 26.55%. Compared with the total *BC*, the *BC* was reduced by 64,800 gha, which accounted for the *BC* with an assumption of 1.75%. The *BC* per capita with this assumption is 0.57 gha, and this value was the same as the *BC* per capita with the matrix analyses transfer of land use.

According to the *ED* formula, we obtained the total *ED* of Kunming City of 17.92 million gha, and the revisions had no effect on the ED which was 2.60 gha per capita. The *ED* change was zero, but it was 5.78 times greater than the per capita *EF* of Yunnan Province. This result was consistent with the GDP per capita.

### 3.4. The Impact of Revised Built-Up Land Factors on the EF and BC Assessment

#### 3.4.1. The Effects of Revisions on Different Research Scales

Comparing the results of *EF* in Yunnan Province and Kunming City, we found that the effects of revisions on different research scales were different. In 2010, the *EF* of Yunnan Province was 88.03 million gha without revisions, while it was 87.82 million gha with revisions. The revised assessment of

Yunnan Province's *EF* was reduced by 0.2 million gha, which accounted for 0.23% of the uncorrected *EF*. Analyzing the components of *EF*, it was found that the correction only affected the *EF* of built-up land, but did not affect the other components. In Figure 4, the *EF* of the built-up land accounted for 1.24% of the total *EF*, which is negligible. Therefore, the impact of corrections on the structure of *EF* is insignificant. The *EF* of built-up land was 1.08 million gha without revisions, while it was 0.88 million gha with revisions. The reduction was 0.2 million gha, which accounted for 18.51% of the uncorrected *EF*. It was equivalent to about a fifth of the built-up land being neglected. Such a large proportion made it impossible to ignore in the calculation. The reason for the reduction was that the yield and equivalence factors of built-up land, with revisions, were smaller than that without revisions. In 2010, the *EF* of Kunming City was 21.62 million gha without revisions, and it was 21.56 million gha with revisions. The reduction was 0.06 million gha, which accounted for 0.28% of the unrevised *EF*. As in the case of Yunnan Province, the correction only affected the *EF* of built-up land, and barely affected the structure of *EF*. The *EF* of the built-up land was 0.24 million gha without revisions, and it was 0.18 million gha with revisions. The reduction was 0.06 million gha, which accounted for 25% of the uncorrected *EF*. It was equivalent to about a quarter of the built-up land not counted in the calculation of *EF*.

The value and proportion of the reduction for Yunnan Province and Kunming City were different. From the perspective of value, the reduction of total and built-up land *EF* in Yunnan Province was greater than that in Kunming City. However, from the perspective of proportion, the reduction of total and built-up land *EF* in Kunming City was larger than that in Yunnan Province. This phenomenon indicates that the impact of revisions on the calculation of *EF* were related to the research scales. To further explore the relationship, we analyzed the land-use composition of the two cases. The proportion of built-up land area to the total area was 3.28% in Kunming City, and that was 0.8% in Yunnan Province. Therefore, the larger the proportion of built-up area, the larger the accuracy of *EF* calculation with revisions.

### 3.4.2. The Effects of Revisions on Different Research Subjects

In the results, the effects of revisions on *EF*, *BC*, and *ED* were different. In 2010, the *ED* in Yunnan Province was 20.6 million gha, and it was 16.8 million gha in Kunming City. In consequence, the revisions had no impact on the calculation of *ED*. The *BC* was 66.24 million gha without revisions, and it was 66.02 million gha with revisions in Yunnan Province. The reduction was 0.22 million gha, which accounted for 0.32% of the uncorrected *BC*. Similar to the analysis results of *EF*, the revisions only affected the *BC* of built-up land, and barely affected the structure of *BC*. The *BC* of built-up land was 1.09 million gha without revisions, and it was 0.88 million gha with revision. The reduction was 0.22 million gha, which accounted for 19.61% of the unrevised *EF* of built-up land. The proportion was equivalent to about one-fifth of built-up land not calculated in the calculation of *BC*. The *BC* was 3.70 million gha without revisions, and it was 3.64 million gha with revisions in Kunming City. The reduction was 0.06 million gha, which accounted for 1.75% of the uncorrected *BC*. The *BC* of built-up land was 0.24 million gha without revisions, and it was 0.18 million gha with revision. The reduction was 0.06 million gha, which accounted for 26.55% of the unrevised *EF* of built-up land. The proportion was equivalent to about one-quarter of built-up land not calculated in the calculation of *BC*.

Compared with the results in the previous section, the revisions made *EF* and *BC* decrease, but the extents of reduction were different. From the perspective of total volume, it was found that the reduction ratio of *EF* was larger than that of *BC* in both Kunming City and Yunnan Province. From the perspective of built-up land, whether it was in Kunming City or Yunnan Province, the reduction ratio of *EF* was larger than that of *BC*. Therefore, the reduction of *EF* was greater than *BC*, and the effect had nothing to do with the research scales.

### 3.4.3. Sensitivity Analysis of *EF* and *BC* Assessments for the Revision

To further explore the impact of built-up land factors on the *EF* and *BC*, we used Kunming City as an example, and calculated the *EF* and *BC* under the different conversion ratios from cropland to built-up land. The ratio was set within 0 to 1, presenting the potential percentage of built-up land converted from other types. The ratio of 1 means that built-up land was completely from cropland. The proportion of built-up land from other land-use types is based on the proportion of forest, pasture, and fishing ground that accounted for their total area in 2010. The results were shown in the Figure 7. In Figure 5, we see that the *EF* and *BC* were positively correlated with the proportion of built-up land from cropland. The $R^2$ was 0.9981, which indicated that the correlation was significant.

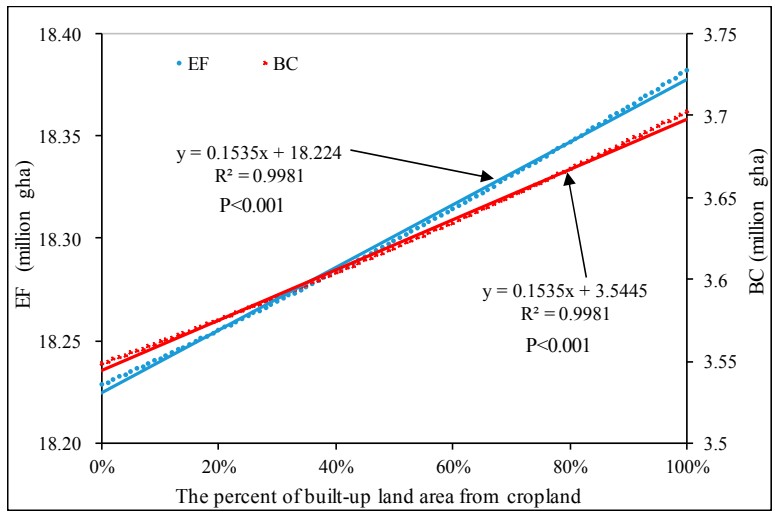

**Figure 7.** The *EF* and *BC* of Kunming City under different percent of built-up land area from cropland.

If the built-up land originated completely from cropland, we calculated that the *EF* was 18.38 million gha, and the *BC* was 3.70 million gha. If the built-up land was from other land-use types that did not contain cropland, then the *EF* was 18.23 million gha, and the *BC* was 3.55 million gha. Therefore, the *EF* and *BC* were reduced by 0.83% and 4.15%, respectively. In 2010, the *EF* of built-up land was 0.24 million gha of Kunming City, and the reduction of revised *EF* was equivalent to 63.85% of it. In other words, only 36.15% of built-up land were calculated. If the study area is highly urbanized, such as in eastern China, the figure of reduction is pretty large. The *BC* of fishing ground was 38 thousand gha, and the reduction of revised *BC* was about four times of it. In other words, we completely ignored the fishing ground in the *BC* calculation without the revisions, which is much less than the reduction. From that perspective, it is significant to revise the accounting method for calculating the *EF* and *BC*. In Figure 7, it can be seen that the growth rate of *EF* is greater than that of *BC* as the proportion of cropland increases. This result indicates that the *EF* is more sensitive to the sources of built-up land, and the sources of built-up land have a greater influence on the *EF* in the study area. The conclusion was identical to the previous section.

The calculation of land-use conversion is based on land uses from two periods, in other words, the source of the current land-use area is related to the previous year. In this study, the land-use data used were in 1980 and 2010, and if different land-use data were used, the accuracy of this calculation could be further improved. There are some uncertainties in calculating land-use conversion from different periods. For example, in the Dianchi Lake Basin of Kunming City, there were some reclaimed cropland from lake in 1970s. In the early 20th century, the urban was built around the lake which was mainly converted from the lake. The land-use data used in this study was from 1980, so the source of this part of built-up land was classified as cropland, but in fact, a significant part of it was derived from water. In the future, it is significant to use different land-use data to calculate the land-use conversion area, which is helpful to improve the accuracy.

## 4. Conclusions

The ecological footprints in Yunnan Province and Kunming City are much higher than their biocapacity, indicating unsustainable development modes in the regions. Under the current industrial structure, they rely heavily on fossil energy and the more developed the economy, the higher the dependence on fossil energy. This scenario is a typical resource and energy consuming economic development model. Therefore, energy is the bottleneck of sustainable economic growth in this region.

The influence of revised factors was undoubtedly related to the land-use composition. The larger the proportion of built-up land is, the greater the influence of the revision has on the *EF* and *BC* calculations. The revision led to a decrease in both *EF* and *BC*, with little effect on the entire structure and per capita, and a negligible effect on the *ED*.

The *EF* and *BC* were positively correlated with the proportion of built-up land converted from cropland, but the sensitivity of *EF* was larger. Therefore, the revision will greatly improve the accuracy of *EF* calculation in areas with a large proportion of built-up land. Extending the revised method to the scale of nation or worldwide, the influence will be small because of the small proportion of built-up land.

In this study, we chose Yunnan Province and Kunming City as cases studies for comparative studies at different scales. In the future, we can choose larger scales, such as national or world, or smaller scales, such as county, to further study the impact of revisions on ecological footprint calculations. This study focuses on revising built-up land parameters, and the other parameters are derived from global values. In the future, it is important to obtain local parameters through NPP, energy, and other methods, which can improve the accuracy of ecological footprint calculations further.

**Author Contributions:** Conceptualization, J.L. and W.G.; Data curation, J.L. and X.Z.; Formal analysis, J.L. and X.Z.; Original Draft Preparation, J.L.; and Review and Editing of Final Manuscript, X.C. and W.G.

**Funding:** Key Program of the National Natural Science Foundation of China (41701631), Natural Foundation for Youth Scholars of Yunnan Province of China (Y0120160068), and the Research of Endangered mechanism and Population Reconstruction of Key Species in Yunnan Plateau Lakes.

**Acknowledgments:** This work was supported by the Key Program of the National Natural Science Foundation of China (41701631), Natural Foundation for Youth Scholars of Yunnan Province of China (Y0120160068), and the Research of Endangered mechanism and Population Reconstruction of Key Species in Yunnan Plateau Lakes.

**Conflicts of Interest:** The authors declare no conflict of interest.

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
