# Peer review of "Revising Yield and Equivalence Factors of Ecological Footprints Based on Land-Use Conversion"

_sustainability, doi:10.3390/su10114107_

Round 1

Reviewer 1 Report

Dear Author(s)

The topic developed is of interest and relevance. The research on trends of urban growth in developing countries is a key issue in the sustainability context. Of fundamental importance here is to enhance the accounting methodology and improve data on which the Ecological Footprint is based. This topic could provide an important contribution to this research area. However, there is still some doubt about the overall quality of your paper to be published in SUSTAINABILITY. It is in no way your methodical approach or your findings and discussion, but alone the yet imperfect style of scientific writing. In this sense, I think the paper could improve if the following remarks are taken into account:

       i.        The wording of the abstract and Introduction section should be improved. The main goal and the methodology of the paper should be accurately formulated. The difference between objective and methodology must clearly indicate. In addition, note that the ‘Abstract’ must be able to stand-alone and the objective and methodology are essential elements of a paper.

      ii.        Furthermore, the object of study should be explicit. As you know, the object of study and the objectives of the paper are different things. An ‘object of study’ refers to the phenomena that function in a particular way in a particular context. By contrast, the ‘objective(s)’ provide an accurate description of the specific action you will take in order to reach the aim(s). This is a very important thing to know what we are reading and why it is significant.

     iii.        The Ecological Footprint account includes fishing grounds; please revise your text (lines 47-49; 96-97…). Please, revise the text in its entirely.

    iv.        Accordingly, in legend of Fig. 1 should say ‘fishing ground’.

     v.         I think that the ‘overshoot’ when is used in Ecological Footprint analysis is a global concept it is not a local concept (line 51).

    vi.        Is ‘water’ included in the methodology? This is confusing. In fact, as you know, in Ecological Footprint analysis the water footprint require a specific treatment. I do not know in what sense is used in the manuscript (e.g. line 115). 

   vii.        As you know, “All manufacturing processes rely to some degree on the use of biocapacity, to provide material inputs and remove wastes at various points in the production chain. Thus all products carry with them an embodied Footprint and international trade flows can be seen as flows of embodied demand for biocapacity. In order to keep track of both the direct and indirect biocapacity needed to support people’s consumption patterns, the Ecological Footprint methodology uses a consumer-based approach; for each land use type, the Ecological Footprint of consumption (EFC) is thus calculated as EFC=EFP+EFI-EFE, where EFPis the Ecological Footprint of production and EFIand EFEare the Footprints embodied in imported and exported commodity flows, respectively” (https://www.footprintnetwork.org/content/images/uploads/National_Footprint_Accounts_Method_Paper_2010.pdf). How was taken this into consideration in your accounts?

  viii.        In the ‘Results and Discussion’ section you describe the results, however I cannot see the discussion. Note that the ‘Discussion’ provides the interpretation of the results in the context of the existing knowledge. In the other hand, the ‘Discussion’ typically ends with implications for theory, future research or possible practical applications of the results. In this sense, you should indicate how the method that you have used allows replications of the research. This is essential in a research paper.

In my opinion, the ‘Conclusion’ section should be improved. Note that the ‘Conclusion’ section –as the ‘Abstract’– is often read in isolation before the other sections. Please, before sending your manuscript again, you may find it helpful to have a colleague that could review it for clarity and accuracy, especially in the sense indicated above.

We would like to thank the referee for taking the time to comment on our manuscript and offering many constructive suggestions. Detailed responses are given below (in italic).

Author Response

Response to Reviewer #1:

1. The wording of the abstract and Introduction section should be improved. The main goal and the methodology of the paper should be accurately formulated. The difference between objective and methodology must clearly indicate. In addition, note that the ‘Abstract’ must be able to stand-alone and the objective and methodology are essential elements of a paper.

Authors’ response: Thank the reviewer for the suggestion. We agree that the “Abstract” is essential which should include objective and method of the paper and must be read in isolation. The objectives of the paper are to evaluate the impact of assumption for calculation under different research scales and enhance the accounting method. The method is based on actual land-use conversion in two various urbanized areas of Yunnan Province and Kunming City from 1980 to 2010 at ArcGIS platform. Via the proportions of different lands converted to built-up land, the parameters of built-up land were calculated by an area-weighting approach. Compared the revised and unrevised EF, the impact of assumption for calculation has been evaluated. To improve the abstract and introduction, we have made some revisions as following.

Ø P1 L11 We have deleted “With the expansion of built-up area from urbanization, the yield and equivalence factors of built-up land are becoming increasingly important in the calculation of ecological footprint (EF).”

Ø P1 L15 We have added “Whereas, the built-up land may be derived from other types of land use. With the expansion of built-up area from urbanization, the yield and equivalence factors of built-up land are becoming increasingly important in the EF calculation.”

Ø P1 L19 We have added ” the proportions of different lands converted to built-up land have been evaluated based on actual land-use conversion in two various urbanized areas of Yunnan Province and Kunming City from 1980 to 2010 at ArcGIS platform.”

Ø P1 L21 We have deleted “the proportions of different lands converted to built-up land in two various urbanized areas of Yunnan Province and Kunming City were obtained, and”

2. Furthermore, the object of study should be explicit. As you know, the object of study and the objectives of the paper are different things. An ‘object of study’ refers to the phenomena that function in a particular way in a particular context. By contrast, the ‘objective(s)’ provide an accurate description of the specific action you will take in order to reach the aim(s). This is a very important thing to know what we are reading and why it is significant.

Authors’ response: Thank the reviewer for the suggestion. We agree that the object of study and the objectives the paper are different things which are very important. In the paper, the objectives are to evaluate the impact of assumption for calculation under different research scales so as to improve the accuracy of the accounting method for ecological footprint. The object of study is the assumption that the built-up land is derived from cropland. To make them clearer, we have made the following revisions.

Ø P2 L70 We have added “Therefore, the aim of the study is to evaluate the impact of the assumption so as to enhance the accounting method”

Ø P2 L72 We have added “take cases of Yunnan Province and Kunming City, and”

Ø P2 L76 We have added “from 1980 to 2010 through ArcGIS platform”

Ø P2 L82 We have added “by comparing the changes in Yunnan Province and Kunming City.”

3. The Ecological Footprint account includes fishing grounds; please revise your text (lines 47-49; 96-97…). Please, revise the text in its entirely.

Authors’ response: Thank the reviewer for the correction. In fact, the “water” represents “fishing ground” in the paper. To avoid misunderstanding, the “water” has been changed to “fishing ground” entirely in the paper.

4. Accordingly, in legend of Fig. 1 should say ‘fishing ground’.

Authors’ response: Thank the reviewer for the correction. In the legend of Fig.1, the “water” has been replaced by “fishing ground”.

5. I think that the ‘overshoot’ when is used in Ecological Footprint analysis is a global concept it is not a local concept (line 51).

Authors’ response: Thank the reviewer for the question. We agreed that the ‘overshoot’ is a global concept. In the paper, the “overshoot” means “ecological deficit (ED)”. If the ecological footprint exceeds the biocapacity of a region, meaning there is an ecological deficit, it implies that a scale mismatch occurs locally and that the region is importing service demand through trade at larger scales. (Liu and Zheng, 2012; Yao et al., 2016)

To avoid misunderstanding, we have changed “EO” to “ED” entirely in the paper.

P6 L196 we have changed “If the EF is larger than the BC, then there is ecological overshoot (EO), indicating that social development and human consumption of natural resources in a specific region has surpassed its ecological carrying capacity.

EO=EF-BC

to “If the BC is larger than the EF, there is ecological reserve (ER). If the EF is larger than the BC, there is ecological deficit (ED), indicating that a mismatch occurs locally in the region [37].

ED=EF-BC”

6. Is ‘water’ included in the methodology? This is confusing. In fact, as you know, in Ecological Footprint analysis the water footprint require a specific treatment. I do not know in what sense is used in the manuscript (e.g. line 115).

Authors’ response: Thank the reviewer for the question. In the paper, the “water” represents “fishing ground” which includes aquatic products. It isn’t the water footprint. To avoid misunderstanding, the “water” has been changed to “fishing ground” entirely in the paper.

7. As you know, “All manufacturing processes rely to some degree on the use of biocapacity, to provide material inputs and remove wastes at various points in the production chain. Thus all products carry with them an embodied Footprint and international trade flows can be seen as flows of embodied demand for biocapacity. In order to keep track of both the direct and indirect biocapacity needed to support people’s consumption patterns, the Ecological Footprint methodology uses a consumer-based approach; for each land use type, the Ecological Footprint of consumption (EFC) is thus calculated as EFC=EFP+EFI-EFE, where EFP is the Ecological Footprint of production and EFI and EFE are the Footprints embodied in imported and exported commodity flows, respectively”(https://www.footprintnetwork.org/content/images/uploads/National_Footprint_Accounts_Method_Paper_2010.pdf). How was taken this into consideration in your accounts?

Authors’ response: Thank the reviewer for the question. In fact, there are two methods to calculate the EF of consumption.

One method:

where EFP is the EF embedded in locally produced products, EFI is the EF of imported or input products, and EFE is the EF of exported or output products.

Another method:

where i is the set of six land use types; N is the total population; Ci is per capita consumption of i; Pi is the total consumption of i; Yni is the yield of i in the region; YFi is the yield factor of i; EQFi is the equivalence factor of i. The method has been adopted in relevant researches (Liu and Zheng, 2012; Yin et al., 2017; Ding and Peng, 2018; Luo et al., 2018).

With the incompleteness of the trade data, the accuracy of the ecological footprint and its components will be reduced. Therefore, we adopted the second method.

To avoid misunderstanding, we have changed “

to “

where i is the set of six land use types; N is the total population; Ci is per capita consumption of i; Pi is the total consumption of i; Yni is the yield of i in the region; YFi is the yield factor of i; EQFi is the equivalence factor of i.

8. In the ‘Results and Discussion’ section you describe the results, however I cannot see the discussion. Note that the ‘Discussion’ provides the interpretation of the results in the context of the existing knowledge. In the other hand, the ‘Discussion’ typically ends with implications for theory, future research or possible practical applications of the results. In this sense, you should indicate how the method that you have used allows replications of the research. This is essential in a research paper.

Authors’ response: Thank the reviewer for the suggestion. We have added the following content to improve the discussion.

Ø P10 L282 We have added “The phenomena is similar to the EF of China from 1978 to 2010 that the proportion of cropland is always the largest component of production footprint [38]. Because China is an agricultural country, and its production and consumption for agriculture are higher than other products. However, the EF of fossil fuel-carbon has become the largest component in consumption footprint since 2005, which had increased to 35% in 2010. The similar proportion of fossil fuel-carbon indicates the fossil fuel consumption of Yunnan Province has reached the average of China. The difference is the proportion of pasture in Yunnan Province is less than that in China since the production and consumption for pasture in Yunnnan Province is scarce”

Ø P11 L311 We have added “Cropland, forest, pasture, fishing ground, and built-up land account for 36.84%, 52.03%, 9.45%, 0.34%, and 1.33% of the BC, respectively. The largest proportion is forest indicating that the forest provides abundant forest productions and absorbs much waste gases in Yunnan Province.”

Ø P12 L354 We have added” Compared with Wuhan City, the proportion of carbon footprint in Wuhan City exceeds 80% [37], much higher than 44.62% in Kunming City. The reason is Wuhan is a heavy industry city. In 2010, the secondary industry added value of Wuhan City was 352.02 billion yuan, 3.39 times of Kunming City of 96.086 billion yuan. Therefore, Wuhan City relies more on energy and source consumption than Kunming City.”

Ø P13 L367 We have added” Cropland, forest, pasture, fishing ground, built-up land account for 40.44%, 40.34%, 11.60%, 1.03%, and 6.59% of the BC, respectively. The largest proportions are cropland and forest which indicates the forest resource is a primary element to the local biocapacity. Compared with Wuhan City, the largest proportion of biocapacity is cropland which exceeds 80%, and the proportion of forest and pasture are lower than 5% [37]. This indicates the ecology supports the biocapacity of Kunming City is much higher than that of Wuhan City. The phenomenon is the same as the function of the two cities in China. Wuhan is a heavy industrial city in central China while Kunming is an eco-city in western China.”

9. In my opinion, the ‘Conclusion’ section should be improved. Note that the ‘Conclusion’ section –as the ‘Abstract’– is often read in isolation before the other sections. Please, before sending your manuscript again, you may find it helpful to have a colleague that could review it for clarity and accuracy, especially in the sense indicated above.

Authors’ response: Thank the reviewer for the suggestion. We agree that the “Conclusion” should be expressed for clarity and accuracy, and some revisions as followings are made to the manuscript.

Ø P14 L414 We have deleted “Using the matrix analysis of land use in Yunnan Province and Kunming City from 1980 to 2010, we revised the yield and equivalence factors of built-up land. Through the EF model, we calculated the EF, BC and EO of Yunnan Province and Kunming City in 2010. In terms of the revised and unrevised EF, BC and EO, the influences of the yield and equivalence factors on these variables are obtained. Finally, using Kunming City as the research object, under the condition that the relative proportion of other land uses are constant, the influence of the proportion of built-up land from cropland on EF and BC is explored. The following conclusions are drawn:”

Ø P14 L420 We have changed “First, the EOs of Yunnan Province and Kunming City are more than zero, and they are unsustainable. The size of the EO is related to the local economy and level of urbanization. Regions with relatively developed economies and higher urbanization rates have larger EO.” to “The ecological footprints in Yunnan Province and Kunming City are much higher than their biocapacity, indicating unsustainable development modes in the regions”

Ø P14 L427 We have changed “Second, the revised factors vary greatly in the different regions, which is related to their industrial structure. The more developed the secondary and tertiary industries, the greater the influence of the revised factors have on the EF and BC in the region. In general, the revised factors decrease. This scenario leads to a decrease in the EF and BC, but this scenario has no effect on the EO. The revised factors significantly reduced the EF and BC of built-up land but had little effect on the entire structure of the EF and BC. When the EF and BC were averaged to per capita, the revised factors have little effect on them.” to “The influence of revised factors is undoubtedly related to the land use composition. The larger the proportion of built-up land is, the greater the influence of the revision have on the EF and BC calculation. The revision leads to a decrease in both EF and BC, little effect on the entire structure and per capita, and negligible effect on the EO.”

Ø P14 L436 We have changed “Finally, the EF and BC are positively correlated with the proportion of built-up land from cropland, but their sensitivity to the proportion is different. The sources of built-up land have greater influence on the EF. Furthermore, the influence is related to the level of the economy, urbanization, policy, etc.” to “The EF and BC are positively correlated with the proportion of built-up land converted from cropland, but the sensitivity of EF is larger. Therefore, the revision will greatly improve the accuracy of EF calculation in areas with a large proportion of built-up land. Extending the revised method to the scale of nation or worldwide, the influence will be little with the reason of small proportion of built-up land.”

Ø P14 L442 We have added “The study focuses on revising built-up land parameters, and the other parameters are derived from global values. In the future, we can obtain local parameters through NPP, energy and other methods, which can improve the accuracy of ecological footprint calculation further.”

References:

Ding, Y. and Peng, J., 2018. Impacts of Urbanization of Mountainous Areas on Resources and Environment: Based on Ecological Footprint Model. Sustainability, 10:765.

Li, X., Tian, M., Wang, H., Wang, H. and Yu, J., 2014. Development of an ecological security evaluation method based on the ecological footprint and application to a typical steppe region in China. Ecological Indicators, 39:153-159.

Liu, J. and Zheng, X., 2012. A Spatial-EF and Econometrics Model Integrated Approach to Explore Land Use Sustainable Forecast Model-in Case of Shandong Province. Procedia Environmental Sciences, 12:413-420.

Luo, W., Bai, H., Jing, Q., Liu, T. and Xu, H., 2018. Urbanization-induced ecological degradation in Midwestern China: An analysis based on an improved ecological footprint model. Resources Conservation & Recycling, 137:113-125.

Monfreda, C., Wackernagel, M. and Deumling, D., 2004. Establishing national natural capital accounts based on detailed Ecological Footprint and biological capacity assessments. Land Use Policy, 21:231-246.

Yao, X., Wang, Z. and Zhang, H., 2016. Dynamic Changes of the Ecological Footprint and Its Component Analysis Response to Land Use in Wuhan, China. Sustainability, 8:329.

Yin, Y., Han, X. and Wu, S., 2017. Spatial and temporal variations in the ecological footprints in northwest China from 2005 to 2014. Sustainability, 9.

Reviewer 2 Report

Dear authors,

General comments

In this paper, authors propose ecological footprint a revised Ecological Footprint (EF) calculations, where the built-up land may be derived from other types of land use in addition to cropland and taking into account the increasingly importance of the yield and equivalence factors of built-up land in the EF calculation. Specifically, the objective of the study is to evaluate the influence of different land uses, as origin land uses of the constructed land and the comparison of the EF calculations considering this multiple origin and without this consideration (only conversion of cropland to built-up land). For this objective, they evaluated land-use conversion in two various urbanized areas of Yunnan Province and Kunming City from 1980 to 2010 at ArcGIS platform. The results showed that in both cases the EFs are greater than their biocapacities (BCs), indicating unsustainable states, and, the values calculated using the unrevised and revised equations of EF and BC differed significantly. They conclude, therefore, it is of great significance to revise the yield and equivalence factors of built-up land using actual land-use conversions in highly urbanized areas to calculate EF and BC.

The EF and BC are considered biophysical indicators of sustainability for environmental management but, from the point of view of the Environmental Economy, their capacity for accounting sustainability is limited, and for this reason they have been considered as indicators of weak sustainability. However, due to the simplicity of their calculations and their value as indicators, studies aimed at improving their reliability are of interest in the field of environmental management. Therefore, I consider that the manuscript (MS) is useful and fits the topics and goals of the journal and may be of interest to its readers. In this corrected version of the MS the authors have responded rigorously and in a satisfactory manner to the comments of previous reviewers. The quality of the manuscript has improved greatly. But, having said that, I also consider that there are two main recommendations to improve the paper: a) the authors must justify the two comparative case studies considering that one corresponds to a province of almost 400,000 km2 and almost 50 million inhabitants and the other corresponds to a city (capital of that province) of little more than 20,000 km2 and 6 million inhabitants; b) the authors should further discuss the benefits of applying the revised values for the calculation of EF and BC, since that is their objective and their novel contribution.

General comments of the MS:

The MS is, in general, clear and concise.

Title and abstract are correct.

Keywords are almost all in the title, some of them could be substituted.

Objectives could be clearer.

Methods are appropriate and repeatable.

Results and discussion are correct but the contribution of the paper should be highlighted. Figures could be improved and captions and figures of tables should be improved. Conclusion section is correct.

Specific comments of the MS:

Page 2; line 70: As it is written, the objective is understood but confusing. I suggest: “Therefore, the aim of the study is to assess the impact of this hypothesis in order to improve the accounting method for calculating the EF and BC” or another alternative that the authors consider.

Page 8; lines 250-254: What do these laws say about? Why the authors consider it to be the main factor?

Page 9; lines 263-264: For me this is key, and almost obvious, in all the results, despite the fact that proportions are compared. The province of Yunnan is so homogeneous for data considered? Probably its capital is the 'only' urban area of the province and the rest could be considered surrounding rural area. This is unclear to me. Please, clarify.

Page 10; Table 2: Please, include separation of thousand units in the figures, include total consumption and emissions in the Table and lack the asterisk of the footnote to the Table (to which part of the table this clarification belongs).

Page 10; line 282: Why however?

Page 10; line 286-287: Wouldn't fossil fuel-carbon contribute the most?

Page 10; lines 302-303: Some discussion referred to the results of these publications? The quotation 48 does not exist, it refers to the 47?

Page 11; Table 3: Please, include separation of thousand units in the figures.

Page 11; line 330: In this sentence: “…of per capita…”; delete of, please.

Page 12; line 353: “This result is strongly related to Kunming's industrial structure”. This fact is key. Authors must discuss about this because, from my point of view, the results depend on this factor and the wide difference in surface and population data between the both cases of study.

Page 12; line 354: “…primary, secondary and tertiary industries…”. In this case industries is equivalent to sectors?

Page 13; line 376-377: “37]. “This indicates the ecology supports the biocapacity of Kunming City is much higher than that of Wuhan City”. I believe that this sentence lacks a conjunction as a nexus (that?) or it is not understood. What the authors refer to with this phrase: "...the ecology supports the biocapacity...". What does ecology mean in this context? Why Kunming City is an eco-city when secondary and tertiary sectors represent almost 95% of its GDP? In fact, their results show: "The ecological footprints in Yunnan Province and Kunming City are much higher than their biocapacity, indicating unsustainable development modes in the regions". It is confusing or contradictory to speak of an eco-city with high ED and unsustainable. Please, clarify.

Author Response

Response to Reviewer #2:

Response to General Comments of the MS:

1. The authors must justify the two comparative case studies considering that one corresponds to a province of almost 400,000 km2 and almost 50 million inhabitants and the other corresponds to a city (capital of that province) of little more than 20,000 km2 and 6 million inhabitants.

Authors’ response: Thank the reviewer for the suggestions. In this paper, we first used the land use transformation matrix method to revise the calculation of EF. The effects of revisions were verified using two cases of Yunnan Province and Kunming City. The impact of revisions on the EF calculation is illustrated by comparing the changes before and after the revision of EF in Yunnan Province and the changes before and after the revision of EF in Kunming City. Compared with Kunming City, Yunnan Province has a larger area and more population. Therefore, Yunnan Province belongs to a relatively large research scale, and Kunming City is a relatively small research scale. The effects of revisions on the EF calculation is further illustrated by comparing the changes at the two scales. Of course, we can choose larger scales, such as nation, world, or smaller scales, such as county towns. Currently, we collect complete data of Yunnan Province and Kunming City. Therefore, they are used as case studies. To make it clearer, we have made the following revisions.

Ø P2 L74 We have changed the sentence into “Based on the integrity and accessibility of data, we chose Yunnan Province and Kunming City which respectively belonged to a relatively large and relatively small research scale as case studies, and calculated their EF and BC in 2010.”

Ø P18 L472 We added “In the study, we chose Yunnan Province and Kunming City as case studies for comparative studies at different scales. In the future, we can choose larger scales, such as nation, world, or smaller scales, such as county, to further study the impact of revisions on ecological footprint calculations.

2. The authors should further discuss the benefits of applying the revised values for the calculation of EF and BC, since that is their objective and their novel contribution.

Authors’ response: Thank the reviewer for the suggestions. To make it clearer, we have made the following revisions.

Ø P17 L429 We added “In 2010, the EF of build-up land was 0.24 million gha of Kunming City and the reduction of revised EF was equivalent to 63.85% of it. In other words, only 36.15% of build-up land has been calculated. If the study area is highly urbanized, such as Beijing, the figure of reduction is pretty large. The BC of fishing ground was 38 thousand gha and the reduction of revised BC was about four times of it. In other words, we completely ignored the fishing ground in the BC calculation without the revisions which is much less than the reduction. From the perspective, it’s significant to revise the accounting method for calculating the EF and BC.”

Response to Specific Comments of the MS:

1. Page 2; line 70: As it is written, the objective is understood but confusing. I suggest: “Therefore, the aim of the study is to assess the impact of this hypothesis in order to improve the accounting method for calculating the EF and BC” or another alternative that the authors consider.

Authors’ response: Thank the reviewer for the correction.

Ø P2 L70 We have changed the sentence into “Therefore, the aim of the study is to assess the impact of this hypothesis in order to improve the accounting method for calculating the EF and BC.”

2. Page 8; lines 250-254: What do these laws say about? Why the authors consider it to be the main factor?

Authors’ response: Thank the reviewer for the questions. The laws on the official website of Municipal Bureau of Land and Resources said about the ordinances of land transfer and management, land-use planning, the regulations of basic farmland protection and so on. These laws projected specific plans of land use in the study area for the next decade. In China, the laws and regulations have great influence on land use change under the planned and market economy. To make it clearer, we have made the following revisions.

Ø P9 L252 We have changed the sentence into “As of December 31, 2010, there were total 93 laws and regulations, related to land transfer, management, and land-use planning, had been showed on the official website of Kunming Municipal Bureau and Yunnan Provincial Department of Land and Resources.”

Ø P9 L258 We have changed the sentence into “The factor was one of the main reasons that led to the difference in the land-use change.”

3. Page 9; lines 263-264: For me this is key, and almost obvious, in all the results, despite the fact that proportions are compared. The province of Yunnan is so homogeneous for data considered? Probably its capital is the 'only' urban area of the province and the rest could be considered surrounding rural area. This is unclear to me. Please, clarify.

Authors’ response: Thank the reviewer for the questions. Yunnan Province has sixteen cities and Kunming City is one of it. There are many urban areas of the province in addition to the capital from the figure 1.

4. Page 10; Table 2: Please, include separation of thousand units in the figures, include total consumption and emissions in the Table and lack the asterisk of the footnote to the Table (to which part of the table this clarification belongs).

Authors’ response: Thank the reviewer for the corrections.

Ø P11 T2 We have separated the figures with thousand units, added total consumption and emissions, and added the asterisk of the footnote to the Table.

Table 2 The EF of fossil fuel-carbon

Category

Amount consumed (t)

Carbon emissions (t)

Fossil energy land (gha)

Coal*

49,208,566.41

35,203,808.41

25,773,154.38

Liquid fossil fuel*

13,106,670.87

9,376,512.34

6,864,663.54

Fossil gas*

485,753.52

347,508.07

254,415.06

Total

62,800,990.8

44,927,828.82

32,892,232.99

*The amount of consumption of fossil has been transformed to standard coal.

5. Page 10; line 282: Why however?

Authors’ response: Thank the reviewer for the question. “However” is inapposite in this position.

Ø P11 L288 We have changed the word to “Hence”.

6. Page 10; line 286-287: Wouldn't fossil fuel-carbon contribute the most?

Authors’ response: Thank the reviewer for the question. In the sentence, we wanted to indicate that the component proportion of EF is similar to China from 1978 to 2010. In fact, the EF of fossil fuel-carbon is the largest from 2005 in China. To make it clearer, we have made the following revisions.

Ø P11 L 292 We have changed the sentence into “The phenomena is similar to the EF of China from 1978 to 2010”

7. Page 10; lines 302-303: Some discussion referred to the results of these publications? The quotation 48 does not exist, it refers to the 47?

Authors’ response: Thank the reviewer for the questions. The sentence means that the proportion of fossil fuel of EF is largest in these publications. This phenomenon is similar to Yunnan province. The quotation 48 does not exist, it refers to the 47. We re-checked the references and renumbered them in the paper. To make it clearer, we have made the following revisions.

Ø P12 L309 We have changed the sentence into “This phenomenon is similar to those in publications”

8. Page 11; Table 3: Please, include separation of thousand units in the figures.

Authors’ response: Thank the reviewer for the correction.

Ø P14 T3 We have separated the figures with thousand units.

Table 3 The BC of Yunnan under different assumptions

Land use

Area (ha)

Revision (gha)

No revision (gha)

Cropland

6,855,811.11

24,323,046.67

24,323,046.67

Forest

22,029,953.08

34,353,508.84

34,353,508.84

Pasture

8,669,372.65

6,241,948.31

6,241,948.31

Fishing ground

287,044.18

225,272.27

225,272.27

Built-up land

308,361.78

879,447.78

1,094,005.91

9. Page 11; line 330: In this sentence: “…of per capita…”; delete of, please.

Authors’ response: Thank the reviewer for the correction.

Ø P14 L337 We have deleted “…of per capita…”.

10. Page 12; line 353: “This result is strongly related to Kunming's industrial structure”. This fact is key. Authors must discuss about this because, from my point of view, the results depend on this factor and the wide difference in surface and population data between the both cases of study.

Authors’ response: Thank the reviewer for the suggestions. In 2010, the proportion of tertiary industry in Kunming City, account for 49.01% of the GDP, is larger than that in Yunnan Province for 40.04% of the GDP. The tourism in tertiary industry provided the largest contribution and there are many people and much consumption with it. This leads to the proportion of fishing ground of EF in Kunming City is larger than that in Yunnan Province. To make it clearer, we have made the following revisions.

Ø P14 L360 We added “In 2010, the proportion of tertiary industry in Kunming City, accounted for 49.01% of GDP, was larger than that in Yunnan Province for 40.04% of GDP. The tourism in tertiary industry provided the largest contribution. This leads to the proportion of fishing ground of EF in Kunming City is larger than that in Yunnan Province.”

11. Page 12; line 354: “…primary, secondary and tertiary industries…” In this case industries is equivalent to sectors?

Authors’ response: Thank the reviewer for the question. In this case, industries are equivalent to sectors. The "industry" we refer to is divided according to the description in China's "National Economic Industry Classification" (GB/T 4754-2011), including the primary industry, the secondary industry and the tertiary industry. It is equivalent to the meaning of sectors.

12. Page 13; line 376-377: “This indicates the ecology supports the biocapacity of Kunming City is much higher than that of Wuhan City”. I believe that this sentence lacks a conjunction as a nexus (that?) or it is not understood. What the authors refer to with this phrase: "...the ecology supports the biocapacity..." What does ecology mean in this context? Why Kunming City is an eco-city when secondary and tertiary sectors represent almost 95% of its GDP? In fact, their results show: "The ecological footprints in Yunnan Province and Kunming City are much higher than their biocapacity, indicating unsustainable development modes in the regions". It is confusing or contradictory to speak of an eco-city with high ED and unsustainable. Please, clarify.

Authors’ response: Thank the reviewer for the questions. In fact, the sentence means that the biocapacity of forest and pasture in Kunming City has larger contribution to the total biocapacity than that in Wuhan City. As the reviewer said, both cities are unsustainable. The sentence wants to indicate the main goal of Wuhan City is heavy industry and the main goal of Kunming City is to protect environment and develop tourism in the urban development planning. To make it clearer, we have made the following revisions.

Ø P16 L387 We have changed the sentence into “the biocapacity of forest and pasture in Kunming City has larger contribution to the total biocapacity than that in Wuhan City”

Ø P16 L389 We have changed the sentence into “This phenomenon is the same as the development goals of two cities in China. The main goal of Wuhan is to develop heavy industry. For Kunming, the main goal of future urban development is to develop tourism based on the protection of the environment.”

Reviewer 3 Report

In the introduction, the authors state that land-use changes were analyzed through remote sensing image data, although in the entire paper there are not presented any remote-sensing image data, or how were they analyzed ( they should be presented at „Materials and Methods”). The authours should explain in what way were useful these remote sensing images, what were the precision of these images (are they accurate) etc.

The EF and BC of build-up land and the land-use conversion matrix are properly described and the results are presented clearly. The authours should also include in the „Results” section a map based on the remote sensing images presenting the land-use changes which were analyzed

Author Response

Response to Reviewer #3:

We would like to thank the referee for taking the time to comment on our manuscript and offering many constructive suggestions. Detailed responses are given below (in italic).

1. “In the introduction, the authors state that land-use changes were analyzed through remote sensing image data, although in the entire paper there are not presented any remote-sensing image data, or how were they analyzed ( they should be presented at „Materials and Methods”). The authors should explain in what way were useful these remote sensing images, what were the precision of these images (are they accurate) etc.”

Authors’ response: Thank the reviewer for the suggestion. The following information is added to the manuscript to describe the source and treatment of the remote sensing images used.

Ø P7 L216 We have changed “The land uses data mainly comes from UCL-Geomatics (Belgium) (http://maps.elie.ucl.ac.be/CCI/viewer/index.php), and the years the data was acquired are 1980 and 2010.” to “The remote sensing image data of land uses type data is mainly obtained from UCL-Geomatics (Belgium) (http://maps.elie.ucl.ac.be/CCI/viewer/index.php) with the spatial resolution of 30 m, and the years the data was acquired are 1980 and 2010 respectively. Land use conversion data can be obtained with the remote sensing image data by ArcGIS platform”

2. The EF and BC of build-up land and the land-use conversion matrix are properly described and the results are presented clearly. The authors should also include in the „Results” section a map based on the remote sensing images presenting the land-use changes which were analyzed.

Authors’ response: Thank the reviewer for the suggestion. We agreed and the remote sensing images presenting the land-use changes were analyzed as following.

Ø P8, we have added “Fig. 2 Land uses of Yunnan Province and Kunming City in 1980(a) and 2010(b)”.

Reviewer 4 Report

p.1

Line 20: “…in two various urbanized areas…” → “…in two urbanized areas…”

Line 26: “…degrees. Because of…” → “…degrees, because of …”

p.2

Line 50: “[8-10]” → The authors should cite more updated references ([8] was published in 2000 and was in Chinese; [9] was published in 1999; [10] was published in 2009).

Line 63: “[12,18-21]” → The authors should cite more updated references.

p.6

Line 158-159: “It has been reported that one-third of the total anthropogenic emissions should be deducted from the calculation [34].” → The authors should cite updated references to calculate the absorption of the anthropogenic emissions of the ocean. For example, IGBP et al. (2013) suggested a quarter of emissions will be absorbed by the ocean (rather than “one-third”, because of the acidification of the ocean) (Lee, 2015).

IGBP; IOC; SCOR, 2013, Ocean Acidification Summary for Policymakers - Third Symposium on the Ocean in a High-CO2 World. International Geosphere-Biosphere Programme, Stockholm, Sweden.

Lee, Y.-J. Land, Carbon and Water Footprints in Taiwan. Environmental Impact Assessment Review 2015, 54, 1-8.

p.7

Line 219: “…the years the data was acquired…” → “…the years the data were acquired…”

p.8

Line 244: “The changes are showed in Fig.2” → “The changes are shown in Fig.3”

p.9

Line 262: “…and are shown in Fig. 3” → “…and are shown in Fig. 4”

p.10

Line 283: “…is shown in Fig. 4” → “…is shown in Fig. 5”

Line 295: “This result show that…” → “This result shows that…”

Line 303: “[31, 44-48]” → there is no [48].

p.12

Line 346: “…in the Fig. 5” → “…in Fig. 6.”

Line 348-349: “Based on these the proportions…” → “Based on these proportions…”

Line 349: “…fossil fuel-carbon is largest…” → “…fossil fuel-carbon is the largest…”

p.13

Line 374: “…the forest resource is a primary element” → why is “forest resource” (40.34%), why not “cropland” (40.44%)?

Line 378-379: “…Kunming is an eco-city in western China” → references?

Line 405: “…in the Fig. 6” → “…in Fig. 7”

p.14

Line 431: “…they relies heavily on…” → “…they rely heavily on…”

Line 438: “…the greater the influence of the revision have on the EF…” → “…the greater the influence of the revision has on the EF…”

Line 440: “…on the EO” → “…on the ED”

Author Response

Response to Reviewer #4:

1. P1 Line 20: “…in two various urbanized areas…” “…in two urbanized areas…”

Line 26: “…degrees. Because of…” “…degrees, because of …”

Ø Authors’ response: Thank the reviewer for the corrections.

Ø P1 L20 We have changed the sentence into “…in two urbanized areas…”

Ø P1 L26 We have changed the sentence into “…degrees, because of …”

2. P2 Line 50: “[8-10]” The authors should cite more updated references ([8] was published in 2000 and was in Chinese; [9] was published in 1999; [10] was published in 2009).

Line 63: “[12,18-21]” The authors should cite more updated references.

Authors’ response: Thank the reviewer for the corrections.

Ø P14 L451 We have changed the reference [8] into “Verhofstadt, E.; Ootegem, L.V.; Defloor, B.; Bleys, B. Linking individuals’ ecological footprint to their subjective well-being. Ecol. Econ. 2016, 127, 80–89.”

Ø P14 L453 We have changed the reference [9] into “Xue, Q.; Song, W.; Zhang, Y.; Fengyun, M. Research Progress in Ecological Carrying Capacity: Implications, Assessment Methods and Current Focus. J. Resour. Ecol. 2017, 8, 514–525.”

Ø P14 L455 We have changed the reference [9] into “Yung-Jaan, L.; Li-Pei, P. Taiwan’s Ecological Footprint (1994–2011). Sustainability 2014, 6, 6170–6187.”

Ø P14 L472 We have changed the reference [18] into “Galli, A.; Giampietro, M.; Goldfinger, S.; Lazarus, E.; Lin, D.; Saltelli, A.; Wackernagel, M.; Müller, F. Questioning the Ecological Footprint. Ecol. Indic. 2016, 69, 224-232.”

3. P6 Line 158-159: “It has been reported that one-third of the total anthropogenic emissions should be deducted from the calculation [34].” The authors should cite updated references to calculate the absorption of the anthropogenic emissions of the ocean. For example, IGBP et al. (2013) suggested a quarter of emissions will be absorbed by the ocean (rather than “one-third”, because of the acidification of the ocean) (Lee, 2015).

IGBP; IOC; SCOR, 2013, Ocean Acidification Summary for Policymakers - Third Symposium on the Ocean in a High-CO2 World. International Geosphere-Biosphere Programme, Stockholm, Sweden.

Lee, Y.-J. Land, Carbon and Water Footprints in Taiwan. Environmental Impact Assessment Review 2015, 54, 1-8.

Authors’ response: Thank the reviewer for the suggestions. For the amount of CO2 absorbed by the ocean, there are many parameters in different researches. We adopted the parameters of IPCC in 2001 which have been cited a lot. As experts said, the parameters in the new references may be more conducive to improving the accuracy of the results. But in the paper, we focused on revising the accounting method for calculating the EF and BC. In future calculations, we will take reviewer’s advices and cite the latest references and parameters.

4. P7 Line 219: “…the years the data was acquired…” “…the years the data were acquired…”

Authors’ response: Thank the reviewer for the corrections.

Ø P8 L222 We have changed the sentence into”…the years the data were acquired…”

5. P8 Line 244: “The changes are showed in Fig.2” “The changes are shown in Fig.3”

Authors’ response: Thank the reviewer for the corrections.

Ø P9 L246 We have changed the sentence into” The changes are shown in Fig.3”

6. P9 Line 262: “…and are shown in Fig. 3” “…and are shown in Fig. 4”

Authors’ response: Thank the reviewer for the corrections.

Ø P10 L268 We have changed the sentence into” …and are shown in Fig. 4”

7. P10 Line 283: “…is shown in Fig. 4” “…is shown in Fig. 5”

Line 295: “This result show that…” “This result shows that…”

Line 303: “[31, 44-48]” there is no [48].

Authors’ response: Thank the reviewer for the corrections.

Ø P11 L289 We have changed the sentence into”…is shown in Fig. 5”

Ø P11 L301 We have changed the sentence into” This result shows that…”

Ø We have updated the references and renumbered them.

8. P12 Line 346: “…in the Fig. 5” “…in Fig. 6.”

Line 348-349: “Based on these the proportions…” “Based on these proportions…”

Line 349: “…fossil fuel-carbon is largest…” “…fossil fuel-carbon is the largest…”

Authors’ response: Thank the reviewer for the corrections.

Ø P14 L353 We have changed the sentence into”…in Fig. 6.”

Ø P14 L355 We have changed the sentence into”Based on these proportions…”

Ø P14 L356 We have changed the sentence into”fossil fuel-carbon is the largest…”

9. P13 Line 374: “…the forest resource is a primary element” why is “forest resource” (40.34%), why not “cropland” (40.44%)?

Line 378-379: “…Kunming is an eco-city in western China” references?

Line 405: “…in the Fig. 6” “…in Fig. 7”

Authors’ response: Thank the reviewer for the questions and corrections.

Ø P15 L385 We have changed the sentence into”… the cropland resource is a primary element…”

Ø P16 L391 In the response to reviewer #1, we have change the sentence into “For Kunming, the main goal of future urban development is to develop tourism based on the protection of the environment.”

Ø P16 L420 We have changed the sentence into”in Fig. 7”

10. P14 Line 431: “…they relies heavily on…” “…they rely heavily on…”

Line 438: “…the greater the influence of the revision have on the EF…” “…the greater the influence of the revision has on the EF…”

Line 440: “…on the EO” “…on the ED”

Authors’ response: Thank the reviewer for the corrections.

Ø P17 L453 We have changed the sentence into”…they rely heavily on…”

Ø P17 L459 We have changed the sentence into”the greater the influence of the revision has on the EF…”

Ø P18 L461 We have changed the sentence into”on the ED”

Round  2

Reviewer 1 Report

Dear Author(s)

Thank you for the effort you have made for trying to improve the original text. However, I still consider that the overall quality of your manuscript needs to improve to be published in SUSTAINABILITY. The wording of the text –in its current form– is not easy to read. The amendments and corrections hinder its review. Nevertheless, he manuscript could improve if the following remarks are taken into consideration:

       i.        It would be appropriate to explain clearly what exactly the review of the ecological footprint, as well as other aspects and factors referred to in the text. The methodology section could be a good place for it.

      ii.        The abbreviation of Ecological Footprint and Biocapacity should be without “s” at the end, in all cases.

     iii.        “National Wealth Indicator System”, “Sustainable Economic Welfare Index”, and “Barometer Sustainability” should appear in uppercase.

    iv.        In lines 54 and 55, remove the parenthesis.

     v.        In the determination of the ecological footprint should be incorporated “fishing ground” (lines 56-58). 

    vi.        There is an erratum in line 165: when it says “trrated” should say “treated”.

   vii.        In my opinion, table 2 should be moved to an appendix.

  viii.        Please, revise the lines 255-262, the sum of 73.05% and 29,65% exceeds 100%.

    ix.        Please revise, the lines 512, 513.

     x.         I can see the results of the paper but you should emphasise the discussion. Specially, the results should be discussed by presenting generalizations that arise from them, by explaining extreme or unexpected observations, or by informing the reader about limitations of the methods. The discussion could also indicate the implications for theory, future research or possible practical applications of results.

I still think that the conclusions could be improved in the same sense as I indicated in my first review.

Author Response

Response to Reviewer #1:

1. It would be appropriate to explain clearly what exactly the review of the ecological footprint, as well as other aspects and factors referred to in the text. The methodology section could be a good place for it.

Authors’ response: Thank the reviewer for the suggestions. We should be appropriate to explain clearly the review of EF. The EF was developed by William Ress in 1992 and revised by Wackernagel in 1996. It addresses the question how much of the biosphere’s regenerative capacity human activities demand. It is the sum of ecosystem services used by humans compared to the biologically productive area that these services occupy. These services include provision of resources, housing, infrastructure and absorption of that population’s waste, using prevailing technology and management practices. The EF is not a measure of total human impact but a proxy for human pressure on ecosystems. So, it captures a necessary condition for sustainability.

To make it clearer, we have made the following revisions.

Ø  P5 L116, we have added the paragraph “The EF was developed by William Ress in 1992 and revised by Wackernagel in 1996 to measure the human use of natural resources and the functions of natural support service for human. It measures the sustainability of a region by estimating the amount of mutually exclusive bio-productive area required to sustain human natural resource consumption and assimilate human-generated waste, and comparing it with the biocapacity of this region [7-10].

2. The abbreviation of Ecological Footprint and Biocapacity should be without “s” at the end, in all cases.

Authors’ response: Thank the reviewer for the corrections.

Ø  P1 L25 we have changed the sentence into “In both cases, the EF of Yunnan Province and Kunming City were greater than their biocapacity (BC), indicating that they were in unsustainable states.”

Ø  P1 L26 we have changed the sentence into “The EF and BC of the two studied cases were reduced to varying degrees.”

Ø  P1 L82 we have changed the sentence into “Based on the integrity and accessibility of data, we chose Yunnan Province and Kunming City which have different research scales and various built-up land percentages, as the case studies and calculated their EF and BC in 2010.”

Ø  P1 L89 we have changed the sentence into “Comparing the revised and unrevised EF and BC, the changes in the EF and BC under different built-up land factors were analyzed, and the impact of the assumptions for EF and BC calculations were explored.

3. National Wealth Indicator System”, “Sustainable Economic Welfare Index”, and “Barometer Sustainability” should appear in uppercase.

Authors’ response: Thank the reviewer for the corrections.

Ø  P1 L44 we have changed the sentence into “Therefore, many international organizations and related researchers have explored methods and indicators to quantify sustainability, such as National Wealth Indicator System, Sustainable Economic Welfare Index, and Barometer Sustainability.”

4. In lines 54 and 55, remove the parenthesis.

Authors’ response: Thank the reviewer for the suggestions.

Ø  P2 L55 we have changed the sentence into “The EF is primarily used to calculate the bioproductive land of a region that is necessary for the maintenance of resource consumption and the absorption of generated waste by human activities.”

5. In the determination of the ecological footprint should be incorporated “fishing ground” (lines 56-58).

Authors’ response: Thank the reviewer for the corrections.

Ø  P2 L57 we have changed the sentence into “The EF determines (1) the area of cropland, pasture, forest, and fishing ground used for the production of agriculture, livestock, forest, and fishery respectively that are consumed by humans.”

6. There is an erratum in line 165: when it says “trrated” should say “treated”.

Authors’ response: Thank the reviewer for the corrections.

Ø  P7 L172 we have changed the sentence into “Since CO2 is mainly absorbed by forests in terrestrial ecosystems, fossil energy lands are treated as forests.”

7. In my opinion, table 2 should be moved to an appendix.

Authors’ response: Thank the reviewer for the suggestions. The specific classification which was used in this study has been published in many researches. We deleted the table 2 and added the references.

Ø  P9 L245 We have changed the sentence into “The specific classification which was used in this study has been published in many researches [37-39].”

Ø  P22 L670 We have added the reference”

37.   Zhang, Z.; Wang, X.; Zhao, X.; Liu, B.; Yi, L.; Zuo, L.; Wen, Q.; Liu, F.; Xu, J.; Hu, S. A 2010 update of national land use/cover database of China at 1:100000 scale using medium spatial resolution satellite images. Remote Sens Environ. 2014, 149, 142-154.

38.   Lai, L.; Huang, X.; Yang, H.; Chuai, X.; Zhang, M.; Zhong, T.; Chen, Z.; Chen, Y.; Wang, X.; Thompson, J.R. Carbon emissions from land-use change and management in China between 1990 and 2010. Science Advances, 2016, 2(11):e1601063.

39.   Zhou, Y.; Xiao, X.; Qin, Y.; Dong, J.; Zhang, G.; Kou, W.; Jin, C.; Wang, J.; Li, X. Mapping paddy rice planting area in rice-wetland coexistent areas through analysis of Landsat 8 OLI and MODIS images. Int. J. Appl. Earth Obs. Geoinf. 2016, 46, 1-12.

8. Please, revise the lines 255-262, the sum of 73.05% and 29.65% exceeds 100%.

Authors’ response: Thank the reviewer for the corrections.

Ø  P9 L268 we have changed the sentence into “These parts of the total conversion area accounted for 26.95% of the total area converted, which is different from the assumption of the construction land footprint (Fig. 3).”

9. Please revise, the lines 512, 513.

Authors’ response: Thank the reviewer for the suggestions.

Ø  P19 L518 we have revised the graph and its title

Fig. 7 The EF and BC of Kunming City under different percent of built-up land area from cropland

10. I can see the results of the paper but you should emphasise the discussion. Specially, the results should be discussed by presenting generalizations that arise from them, by explaining extreme or unexpected observations, or by informing the reader about limitations of the methods. The discussion could also indicate the implications for theory, future research or possible practical applications of results.

Authors’ response: Thank the reviewer for the suggestions. As the reviewer said, there were some limitations in the method which we should discuss. Land use data is the key element in this corrections. As we know, the calculation of land use conversion is based on land uses from two periods, in other words, the source of the current land use area is related to the previous year used. In this study, the land use data used were in 1980 and 2010, and if land use data from other periods were used, the results of this calculation may be different. Therefore, there are some uncertainties in calculating land use conversion from different periods. For example, in the Dianchi Lake Basin of Kunming City, there were some reclaimed cropland from lake in 1970s. In the early 20th century, the urban was built around the lake, which was converted from the lake. The land use data used in this study was from 1980 which was later than 1970s, so the source of this part of built-up land was classified as cropland, but in fact, it was derived from water. In the future, it is significant to use different land use data to calculate the land use conversion area, which is conducive to improving the accuracy.

To make it clearer, we have made the following revisions.

P20 L539, we have added the paragraph “The calculation of land use conversion is based on land uses from two periods, in other words, the source of the current land use area is related to the previous year. In this study, the land use data used were in 1980 and 2010, and if different land use data were used, the accuracy of this calculation could be further improved. There are some uncertainties in calculating land use conversion from different periods. For example, in the Dianchi Lake Basin of Kunming City, there were some reclaimed cropland from lake in 1970s. In the early 20th century, the urban was built around the lake which was mainly converted from the lake. The land use data used in this study was from 1980, so the source of this part of built-up land was classified as cropland, but in fact, a significant part of it was derived from water. In the future, it is significant to use different land use data to calculate the land use conversion area, which is helpful to improve the accuracy.”

Reviewer 2 Report

Dear Authors,

I have reviewed the corrected version of the manuscript and the authors' responses to the reviewers, including the response to my previous suggestions.. The authors have responded very rigorously and in a very satisfactory manner to their comments. The quality of the manuscript has been significantly improved and, from my point of view, now warrants publication in the SI of the Sustainability Journal.

A very little observation: Line 284, Page 11 I think that is: This factor (or These factors)...not The factor...

Kind regards

Author Response

Response to Reviewer #2:

I have reviewed the corrected version of the manuscript and the authors' responses to the reviewers, including the response to my previous suggestions. The authors have responded very rigorously and in a very satisfactory manner to their comments. The quality of the manuscript has been significantly improved and, from my point of view, now warrants publication in the SI of the Sustainability Journal.

A very little observation:

Line 284, Page 11 I think that is: This factor (or These factors)...not The factor...

Authors’ response: Thank the reviewer for the suggestions again. We are very grateful for his/her comments that are important to improve our manuscript.

P10 L 290 We have changed the sentence into “This factor was one of the main reasons that led to the difference in the land-use change.”

Reviewer 3 Report

The article has been greatly improved. The authors took into account the observations made by the reviewers.

Author Response

Response to Reviewer #3:

The article has been greatly improved. The authors took into account the observations made by the reviewers.

Authors’ response:

Thanks to the reviewer for detailed comments and constructive suggestions. With the comments, we furtherly discussed the significance of the method and highlighted the meaning of the article. This is conducive to improve the article.

Reviewer 4 Report

p.2

Line 48 “To the quantitative measurement…” → “The quantitative measurement…”

Line 90 “the impact rule of the assumptions…” → meaning unclear.

p. 7

Line 165-166 “are trrated as forests” → meaning unclear.

p.9

Line 240 “in the table 2” → “in Table 2”

Line 243 “performed layer overlay analysis…” → “performed layer overlay analyses…”

Line 244-245 “performed matrix analysis…” → “performed matrix analyses…”

Line 247: “Shrub: Lands covered by trees less than 2 meters high, the canopy cover >40%” → please double-check the description of this item.

Line 247 “Pasture: Grassland with canopy cover between 5-20%” → why “5-20%”?

p.15

Line 368: “the ED per capita is 0.48 gha with revision and without revision” → meaning unclear.

Line 395-396 “…larger than 40.04% of GDP in Yunnan Province for.” → meaning unclear.

p.17

Line 437-438 “and the ED per capita was 2.78 gha with the revision and without the revision” → meaning unclear (cf. Line 368).

Line 442-443 “we found that the effects of…was different” → “we found that the effects of…were different”

Line 447 “In figure 4…” → “In Figure 4”

Line 463-464 “it in Kunming City was larger than…” → meaning unclear.

p.18

Line 508 “…shown in the Fig. 7. In figure 5…” → “…shown in Fig. 7. In Figure 5…”

p.19

Line 524 “In figure 7” → “In Figure 7”

p.20

Line 562 “In the study, we…” → “In this study, we…”

Line 565 “The study focuses…” → “This study focuses…”

p.21

Line 596 “Yung-Jaan, L.; Li-Pei, P.” → “Lee, Y.J.; Peng, L.P.”

Author Response

Response to Reviewer #4:

1. P2

Line 48 “To the quantitative measurement…” The quantitative measurement…”

Line 90 “the impact rule of the assumptions…” meaning unclear.

Authors’ response: Thank the reviewer for the suggestions.

Ø  P2 L49 We have changed the sentence into ”How to quantify sustainability has long been a difficult problem in sustainable development research.”

Ø  P2 L90 We have changed the sentence into ”Furthermore, the impact of the assumptions was explored under different spatial scales by comparing the changes in Yunnan Province and Kunming City.”

2. P7 Line 165-166 “are trrated as forests” meaning unclear.

Authors’ response: Thank the reviewer for the suggestions.

Ø  P2 L171 We have changed the sentence into ”Because CO2 is mainly absorbed by forests in terrestrial ecosystems, fossil energy lands are treated as forests.”

3. P9

Line 240 “in the table 2” in Table 2

Line 243 “performed layer overlay analysis…” performed layer overlay analyses…”

Line 244-245 “performed matrix analysis…” performed matrix analyses…”

Line 247: “Shrub: Lands covered by trees less than 2 meters high, the canopy cover >40%” please double-check the description of this item.

Line 247 “Pasture: Grassland with canopy cover between 5-20%” why 5-20%?

Authors’ response: Thank the reviewer for the suggestions. The description of the land use have been checked in the website of http://www.resdc.cn/data.aspx?DATAID=99, accessed on 3 Nov 2018. The specific classification which was used in this study has been published in many researches. We deleted table 2 and added the references.

Ø  P9 L247 We have changed the sentence into ”The specific classification was shown in the appendix A.”

P9 L254 we have deleted “Table2 The classification system of land use and its definition.”

P23 L705 we have added an appendix A “The classification system of land use and its definition.”

Ø  P9 L250 We have changed the sentence into ”secondly, performed layer overlay analyses”

Ø  P9 L251 We have changed the sentence into ”and fourthly, performed matrix analyses for the obtained data through Excel2016”

Ø  P9 L245 We have changed the sentence into “The specific classification which was used in this study has been published in many researches [37-39].”

Ø  P22 L670 We have added the reference”

37.   Zhang, Z.; Wang, X.; Zhao, X.; Liu, B.; Yi, L.; Zuo, L.; Wen, Q.; Liu, F.; Xu, J.; Hu, S. A 2010 update of national land use/cover database of China at 1:100000 scale using medium spatial resolution satellite images. Remote Sens Environ. 2014, 149, 142-154.

38.   Lai, L.; Huang, X.; Yang, H.; Chuai, X.; Zhang, M.; Zhong, T.; Chen, Z.; Chen, Y.; Wang, X.; Thompson, J.R. Carbon emissions from land-use change and management in China between 1990 and 2010. Science Advances, 2016, 2(11):e1601063.

39.   Zhou, Y.; Xiao, X.; Qin, Y.; Dong, J.; Zhang, G.; Kou, W.; Jin, C.; Wang, J.; Li, X. Mapping paddy rice planting area in rice-wetland coexistent areas through analysis of Landsat 8 OLI and MODIS images. Int. J. Appl. Earth Obs. Geoinf. 2016, 46, 1-12.

3. P15

Line 368: “the ED per capita is 0.48 gha with revision and without revision” meaning unclear.

Line 395-396 “…larger than 40.04% of GDP in Yunnan Province for.” meaning unclear.

Authors’ response: Thank the reviewer for the suggestions.

Ø  P15 L374 We have changed the sentence into ”the revisions had no effect on the ED which was 0.48 gha per capita.”

Ø  P16 L402 We have changed the sentence into ”In 2010, the proportion of tertiary industry in Kunming City accounted for 49.01% of GDP, larger than that in Yunnan Province which accounts for 40.04% of GDP.”

4. P17

Line 437-438 “and the ED per capita was 2.78 gha with the revision and without the revision” meaning unclear (cf. Line 368).

Line 442-443 “we found that the effects of…was different” we found that the effects ofwere different

Line 447 “In figure 4…” In Figure 4

Line 463-464 “it in Kunming City was larger than…” meaning unclear.

Authors’ response: Thank the reviewer for the suggestions.

Ø  P17 L443 We have changed the sentence into “the revisions had no effect on the ED which was 2.60 gha per capita.”

Ø  P17 L448 We have changed the sentence into “we found that the effects of revisions on different research scales were different.”

Ø  P17 L453 We have changed the sentence into “In Figure 4,”

Ø  P17 L469 We have changed the sentence into “But, from the perspective of proportion, the reduction of total and built-up land EF in Kunming City was larger than that in Yunnan Province.”

5. P18

Line 508 “…shown in the Fig. 7. In figure 5…” “…shown in Fig. 7. In Figure 5…”

Authors’ response: Thank the reviewer for the suggestions.

Ø  P18 L515 We have changed the sentence into “In Figure 5, we see that the EF and BC were positively correlated with the proportion of built-up land from cropland.

6. P19

Line 524 “In figure 7” In Figure 7

Authors’ response: Thank the reviewer for the suggestions.

Ø  P19 L533 We have changed the sentence into “In Figure 7, it can be seen that the growth rate of EF is greater than that of BC as the proportion of cropland increases.

7. P20

Line 562 “In the study, we…” In this study, we…”

Line 565 “The study focuses…” This study focuses…”

Authors’ response: Thank the reviewer for the suggestions.

Ø  P20 L571 We have changed the sentence into “In this study, we chose Yunnan Province and Kunming City as case studies for comparative studies at different scales.”

Ø  P20 L574 We have changed the sentence into “This study focuses on revising built-up land parameters, and the other parameters are derived from global values.”

8. P21

Line 596 “Yung-Jaan, L.; Li-Pei, P.” Lee, Y.J.; Peng, L.P.”

Authors’ response: Thank the reviewer for the suggestions.

P21 L605 We have changed the reference into “10. Lee, Y. J.; Peng, L. P. Taiwan’s Ecological Footprint (1994–2011). Sustainability 2014, 6, 6170–6187.”
